# CORAL: ORDER-AGNOSTIC LANGUAGE MODELING FOR EFFICIENT ITERATIVE REFINEMENT

## ABSTRACT

Iterative refinement has emerged as an effective paradigm for enhancing the capabilities of large language models (LLMs) on complex tasks. However, existing approaches typically implement iterative refinement at the application or prompting level, relying on autoregressive (AR) modeling. The sequential token generation in AR models can lead to high inference latency. To overcome these challenges, we propose Context-Wise Order-Agnostic Language Modeling (COrAL), which incorporates iterative refinement directly into the LLM architecture while maintaining computational efficiency. Our approach models multiple token dependencies within manageable context windows, enabling the model to perform iterative refinement internally during the generation process. Leveraging the order-agnostic nature of COrAL, we introduce sliding blockwise order-agnostic decoding, which performs multi-token forward prediction and backward reconstruction within context windows. This allows the model to iteratively refine its outputs in parallel in the sliding block, effectively capturing diverse dependencies without the high inference cost of sequential generation. Empirical evaluations on reasoning tasks demonstrate that COrAL improves performance and inference speed, respectively, achieving absolute accuracy gains of $4.6\%$ on GSM8K and $4.0\%$ on LogiQA, along with inference speedups of up to $3.9\times$ over next-token baselines. Preliminary results on code generation indicate a drop in pass rates due to inconsistencies in order-agnostic outputs, highlighting the inherent quality–speed trade-off.

## 1 INTRODUCTION

Large Language Models (LLMs) have recently achieved remarkable success across a wide range of tasks (Brown et al., 2020; Touvron et al., 2023; OpenAI, 2023; Dubey et al., 2024), such as mathematical problem-solving, logical reasoning, and programming (Yu et al., 2024; Pan et al., 2023; Schick et al., 2023; Rozière et al., 2023). Strategies that enable LLMs to learn from previous mistakes and iteratively refine their outputs have been particularly effective, achieving human-level performance and transforming both academic research and industrial applications (Pan et al., 2024; Ye et al., 2024; OpenAI, 2024). These iterative refinement approaches incorporate feedback—either external or internal—as supervision signals during training (Zelikman et al., 2022; Huang et al., 2023; Shinn et al., 2023; Lightman et al., 2024; Xie et al., 2024), or by developing prompting frameworks that guide the model toward improved generations through methods like guided search or self-refine (Yao et al., 2023; Xie et al., 2023; Madaan et al., 2023).

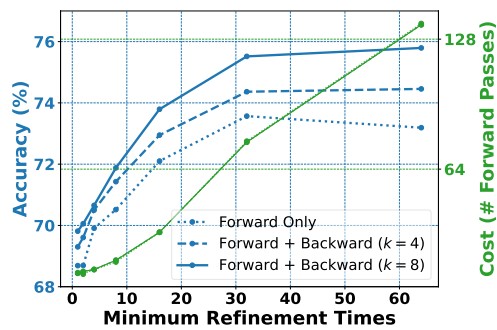

Figure 1: Scaling of performance and inference cost on GSM8K with increasing the minimum refinement times for each output position. $k$ represents the backward context window size. We set the decoding block size as $b = 64$.

Despite their effectiveness, these approaches predominantly rely on autoregressive (AR) LLMs, which generate text by predicting the next token in a fixed left-to-right order using causally masked Transformers (Radford, 2018). This sequential generation process inherently limits the model's

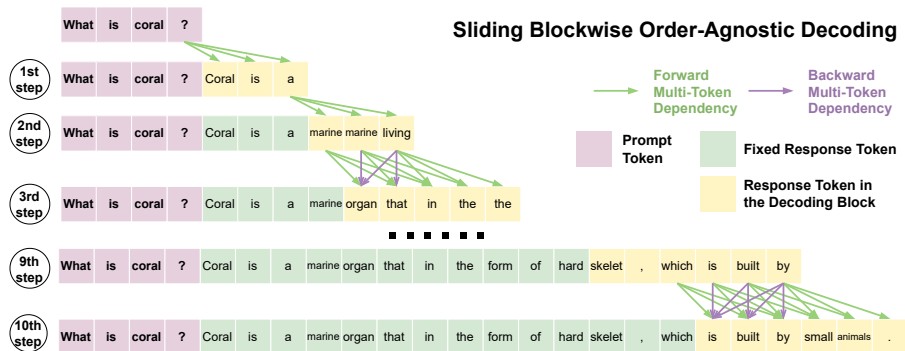

Figure 2: **Sliding Blockwise Order-Agnostic Decoding**. COrAL performs multi-token prediction and refinement in the sliding block with context window size $k=3$ and block size $b=6$.

ability to capture dependencies spanning beyond the immediate next token, especially those that require backward context (Hu et al., 2024). Moreover, the sequential nature of AR models leads to high inference latency, resulting in computational inefficiency for long sequences (Cai et al., 2024).

To address these limitations, researchers have explored order-agnostic architectures that enhance representation learning and accelerate inference. Previous studies mainly focus on two solutions: permutation-based AR and non-autoregressive (NAR) modeling, but each has its own strengths and limitations. For instance, permutation-based models propose diversity-enhanced pretraining objectives that predict multiple subsequent tokens in various orders to capture richer dependencies (Yang et al., 2019; Zhang et al., 2024b). Similarly, NAR models generate tokens in parallel, significantly reducing inference time (Gu et al., 2018). However, conventional NAR models often struggle with tasks involving variable-length generation and complex token dependencies, leading to degraded text quality. As a result, these models are typically task-specific and require additional mechanisms to ensure consistency (Gui et al., 2023; Shi et al., 2024). Inspired by the success of diffusion models in image generation (Austin et al., 2021), recent efforts have adapted denoising techniques to generative language modeling as an iterative extension of NAR models (Savinov et al., 2022; Gong et al., 2023). While these methods improve efficiency, they still lag behind AR models regarding generation quality and generalizability. Given the trade-offs among different models[1], a pivotal question arises:*Can we unify the strengths of denoising techniques with order-agnostic modeling to enhance the capabilities of AR-LLMs while mitigating their respective limitations*?

In this work, we propose **Context-Wise Order-Agnostic Language Modeling** (COrAL), which combines the advantages of AR and order-agnostic modeling. COrAL models token dependencies within manageable context windows, effectively balancing the capture of both local and long-range dependencies with computational efficiency. Through context-wise modeling, COrAL overcomes the limitations of fixed-order generation in AR models and the dependency modeling challenges in NAR models. Within each context window, COrAL models diverse dependencies in an order-agnostic manner, enhancing the model's ability to capture complex token relationships while maintaining computational efficiency. Leveraging COrAL, we introduce **Sliding Blockwise Order-Agnostic Decoding**, which performs forward multi-token prediction and backward reconstruction simultaneously. As shown in Figure 1, this strategy enables the model to perform iterative refinement internally to scale up inference performance. Additionally, to ensure that the model remains aware of target token positions without necessitating architectural changes, we apply a generalized Rotary Position Embedding (RoPE) (Su et al., 2024) to the last layer of the Transformer. This positional encoding technique preserves target-aware representations, which are essential for effective order-agnostic generation and iterative refinement.

With a two-stage training strategy, we equip conventional AR-LLMs with order-agnostic capabilities without requiring architectural add-ons or pre-training from scratch. We conduct extensive experiments on reasoning tasks, including arithmetic computation and logical reasoning, to evaluate the effectiveness and efficiency of COrAL. Our empirical results show that COrAL not only improves performance but also significantly accelerates inference. Specifically, COrAL achieves absolute accuracy gains of $4.6\%$ on GSM8K and $4.0\%$ on LogiQA, along with inference speedups of up to

---

[1]We make conceptual comparison among different model architectures in Appendix A.

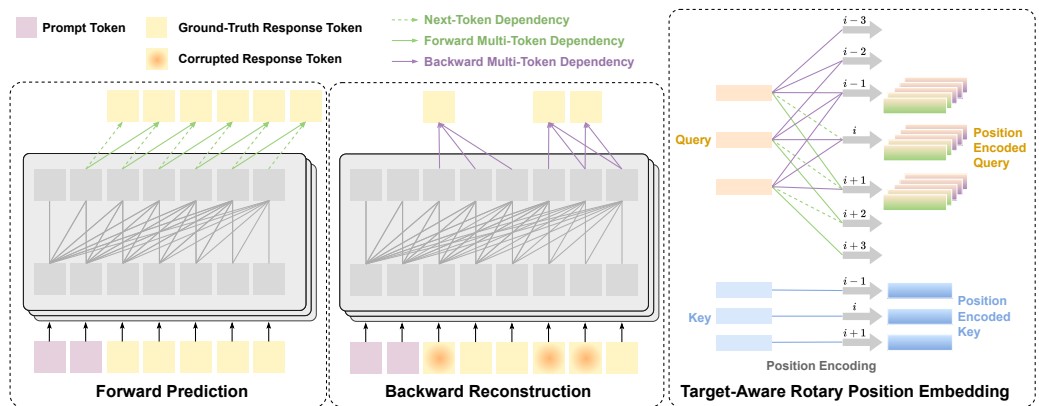

Figure 3: **Context-Wise Order-Agnostic Language Modeling**. We visualize the order-agnostic dependencies within a context window size $k = 2$. For target-aware position encoding, we show how COrAL obtains query representations for multiple positions within a context window size $k = 2$.

3.9 times over next-token baselines. These findings demonstrate that COrAL effectively captures dependencies within context windows while maintaining computational efficiency. However, preliminary experiments on code generation reveal a decrease in pass rates due to inconsistencies in order-agnostic outputs, highlighting the inherent quality–speed trade-offs. This suggests that further refinements are necessary for tasks that require strict syntactic coherence.

## 2 CONTEXT-WISE ORDER-AGNOSTIC LANGUAGE MODELING

We present Context-Wise Order-Agnostic Language Modeling, a generalized AR framework that captures conditional textual distributions based on various orders in context windows.

### 2.1 BACKGROUND

Given a prompt $\boldsymbol{x}$ and a target sequence of $T$ tokens $\boldsymbol{y} = \{y_1, y_2, \cdots, y_T\}$, conventional AR models factorize the multivariate distribution $p(\boldsymbol{y} \mid \boldsymbol{x})$ into a product of univariate distributions using the probability chain rule $\log p(\boldsymbol{y} \mid \boldsymbol{x}) = \sum_{t=1}^{T} \log p(y_t \mid \boldsymbol{y}_{<t}, \boldsymbol{x})$, which requires $T$ iterative sampling steps to generate the sequence. In contrast, order-agnostic AR modeling generalizes this by modeling multiple possible orderings $\sigma \in \mathcal{S}_T$ of the sequence:

$$\log p(\boldsymbol{y} \mid \boldsymbol{x}) = \log \mathbb{E}_{\sigma \sim \mathcal{S}_T} \left[ p(\boldsymbol{y} \mid \boldsymbol{x}, \sigma) \right]$$

$$\geq \mathbb{E}_{\sigma \sim \mathcal{S}_T} \left[ \log p(\boldsymbol{y} \mid \boldsymbol{x}, \sigma) \right] = \mathbb{E}_{\sigma \sim \mathcal{S}_T} \left[ \sum_{t=1}^{T} \log p\left( y_{\sigma(t)} \mid \boldsymbol{y}_{\sigma(<t)}, \boldsymbol{x} \right) \right]. \quad (1)$$

where $\mathcal{S}_T$ denotes the set of all possible permutations of the indices $\{1, 2, \cdots, T\}$. However, this permutation-based objective poses a significant optimization challenge and can lead to underfitting, as observed in prior works (Yang et al., 2019; Hoogeboom et al., 2022).

On the other hand, NAR modeling (Lee et al., 2018) breaks the sequential dependency to accelerate inference. This approach applies sequence-level denoising steps, enabling parallel reconstruction of multiple tokens with iterative refinement to enhance generation quality. To equip the model with denoising capabilities, it employs $L$ intermediate latent variables $\{\boldsymbol{y}^{(1)}, \boldsymbol{y}^{(2)}, \cdots, \boldsymbol{y}^{(L)}\}$ and approximates their marginalization as follows:

$$\log p(\boldsymbol{y} \mid \boldsymbol{x}) \geq \sum_{t=1}^{T} \log p(y_t \mid \boldsymbol{y}^{(L)}, \boldsymbol{x}) + \sum_{l=1}^{L} \sum_{t=1}^{T} \log p(y_t^{(l)} \mid \boldsymbol{y}^{(l-1)}, \boldsymbol{x}) + \sum_{t=1}^{T} \log p(y_t^{(0)} \mid \boldsymbol{x}) \quad (2)$$

where the latent variables are constrained to match the type of the target output $\boldsymbol{y}$. While previous studies demonstrate the efficiency of NAR modeling in specific tasks such as machine translation (Gu et al., 2018; Ghazvininejad et al., 2019; Kasai et al., 2020), its potential in language modeling remains underexplored. Moreover, the use of corrupted data for denoising and the assumption of token-wise independence in each reconstruction step in NAR models can introduce instability, often resulting in reduced text quality compared to their AR counterparts (Savinov et al., 2022).

## 2.2 OBJECTIVE: CONTEXT-WISE ORDER-AGNOSTIC AUTOREGRESSIVE MODELING

To address the above limitations in AR language modeling, we propose Context-Wise Order-Agnostic Language Modeling (COrAL), unifying token-level dependency modeling and sequence-level denoising to advance the capabilities of current LLMs. Previous order-agnostic modeling works attempt to capture various factorization orders involving long dependencies that are difficult to model and fit. In contrast, COrAL learns the orderless relationships within predetermined context windows. Built on the AR foundation, our COrAL framework leverages the superior capability of sequential language modeling in LLMs.

COrAL tackles the problem of generative language modeling by combining forward multi-token prediction with backward denoising in a context-wise and order-agnostic framework. Denoting the context window size as $k$[2], we model the conditional probability distribution of each target token by considering an ensemble of dependencies over all possible positions in the context:

$$\log p_\theta(\boldsymbol{y} \mid \boldsymbol{x}) \geq \sum_{t=1}^{T} \mathbb{E}_{i \sim \mathcal{U}[t-k,t+k]} \mathbb{E}_{l \geq 0} \left[ \log p_\theta(y_t \mid \boldsymbol{y}_{\leq i}^{(l)}, \boldsymbol{x}) \right] \tag{3}$$

where $\boldsymbol{y}^{(l)}$ represents an intermediate state of the target output sequence $\boldsymbol{y}$ during iterative refinement. The conventional AR modeling, in comparison, becomes a specific case where only the forward prediction with $k=1$, conditioned on previous tokens in the target sequence $\boldsymbol{y}$, is modeled.

**Forward Prediction and Backward Reconstruction.** As shown in Figure 3, we decompose the order-agnostic objective into **forward prediction** and **backward reconstruction**. In forward prediction, COrAL learns to predict multiple future tokens simultaneously given past tokens in the ground-truth sequence. For backward reconstruction, we randomly corrupt tokens in the input sequence to create the intermediate states $\boldsymbol{y}^{(l)}$ in Eq. 3. Similar to BERT (Devlin et al., 2019), we compute the loss only on the corrupted tokens. During training, we use the original data for prediction and the corrupted data for reconstruction. This decomposition disentangles the self-refinement capability from forward prediction, leveraging all data points to enhance sequence modeling.

**Corruption Strategy.** Our corruption and reconstruction process is a form of denoising autoencoding (Vincent et al., 2008) in language modeling. However, instead of representation learning, we aim to endow the model with the self-refinement capability to revise the generated content. Inspired by masked autoencoders (He et al., 2022), we divide the output sequence into non-overlapping patches and randomly sample a subset for corruption. Each patch is a fragment of text containing one or multiple consecutive tokens in the sequence. Specifically, we corrupt a patch by either (**i**) replacing it with a random patch sampled from the current sequence or (**ii**) repeating the first token to replace the other tokens in the patch. This design draws on insight from Ye et al. (2024) that model performance can be significantly improved by simply enhancing consistency across steps.

## 2.3 ARCHITECTURE: TARGET-AWARE QUERY REPRESENTATION FOR SELF-ATTENTION

We build our framework by adapting the standard architecture of LLMs using decoder-only Transformers (Brown et al., 2020). Unlike prior NAR works employing encoder–decoder architectures (Lee et al., 2018; Kasai et al., 2020), the conventional AR foundation predicts the same distribution given the current context regardless of the target token position. While this demonstrates advanced capabilities of sequence modeling and generation, the typical parameterization of next-token distribution constrains its generalizability to the order-agnostic objective in Eq. 3.

Previous works on order-agnostic modeling have explored various ways to incorporate positional information, including scaling up the dimensionality of the final projection layer (Stern et al., 2018) and adding look-ahead tokens (Monea et al., 2023) or extra decoding heads (Cai et al., 2024; Gloeckle et al., 2024). Despite their promising performance, these methods introduce the overhead of additional self-attention network calls and new parameters for multi-position prediction. Instead, we propose a seamless adjustment without adding extra model parameters. Specifically, we apply a generalized Rotary Position Embedding (RoPE) (Su et al., 2024) at the final layer of the decoder-only Transformers to integrate target-aware information into the query representations.

---

[2]Without loss of generality, we can set different context window sizes for forward prediction and backward reconstruction in practice. Here, we present the objective with the same hyperparameter $k$ to avoid clutter.

**Target-Aware RoPE.** RoPE encodes positional information into query and key representations, ensuring that their inner product inherently contains relative position information in self-attention: $f(\boldsymbol{q}_m, m)^\top f(\boldsymbol{k}_n, n) = g(\boldsymbol{q}_m, \boldsymbol{k}_n, m-n)$, where $f$ is the positional encoding function applied to the query and key embeddings at $m$-th and $n$-th positions, respectively. Conventional RoPE integrates positional information of the current token to form the query representation. While this effectively enhances the position-aware representation of the input token in intermediate hidden states, it introduces inherent misalignment with the target token position when using the learned representation for output prediction. To avoid this problem, we propose Target-Aware RoPE (Figure 3), which modifies the positional encoding function at the final layer by considering the target token position in the query representation:

$$f(\boldsymbol{q}_m, \mu)^\top f(\boldsymbol{k}_n, n) = g(\boldsymbol{q}_m, \boldsymbol{k}_n, \mu - n), \quad \mu \in [m - k, m + k] \tag{4}$$

The rationale behind this modification is that the position encoding in RoPE can adapt the representation of the current token to be tailored for the target position. This simple yet effective adjustment endows the model with the target-aware capability, allowing it to predict tokens at various positions without the overhead of additional entire network calls.

## 3 Sliding Blockwise Order-Agnostic Decoding

Leveraging the order-agnostic capabilities of COrAL, we propose Sliding Blockwise Order-Agnostic Decoding, a parallel decoding strategy to enable efficient iterative refinement.

High inference latency significantly hinders the broader application of AR-LLMs. Recent studies have tackled this bottleneck from various angles to accelerate inference. For instance, speculative decoding employs a smaller, faster draft model to propose multiple continuations, which the larger target model then verifies and accepts (Leviathan et al., 2023; Miao et al., 2024). Blockwise parallel decoding directly leverages the large model to generate multiple tokens simutaneously (Stern et al., 2018; Cai et al., 2024). However, these studies increase memory consumption, which thus limits the scalability and impedes distributional deployment. Another promising line of work breaks the sequential dependency by adopting Jacobi decoding (Santilli et al., 2023; Fu et al., 2024) for iterative refinement without architectural add-ons. Kou et al. (2024) propose consistency LLMs to further improve the performance of Jacobi decoding inspired by consistency models (Song et al., 2023).

While these existing approaches improve inference efficiency, they rely on the conventional left-to-right AR foundation with monotonic dependencies. In this work, we leverage the order-agnostic nature of COrAL to perform backward sequence-level refinement and forward multi-token prediction simultaneously, significantly accelerating inference. At each step, we ensemble the output distributions based on multiple possible dependencies and construct a candidate set to fill a block of the output sequence. Furthermore, this process facilitates self-refinement by modifying previous generations at a higher-level horizon, enhancing output quality with advanced inference capabilities. Next, we detail the ensemble strategy in decoding for candidate construction and verification, corresponding to the "Collect" and "Verify and Slide" parts in Algorithm 1, respectively.

**Prediction.** Given a set of possible distributions $\{p_\theta(y_t \mid \boldsymbol{y}_{\leq i}, \boldsymbol{x})\}_{i=t-k}^{t+k}$ for the $t$-th token in the output sequence, we obtain the ensemble distribution via model arithmetic (Dekoninck et al., 2024). Specifically, we apply different weights to the distributions to prioritize the more accurate dependencies, with distributions based on more qualified content generally leading to better generations:

$$\pi_\theta(y_t) = \mathrm{softmax}\left(\frac{1}{\sum_{i=t-k}^{t+k} \omega_{t-i}(\boldsymbol{y}_{\leq i}, \boldsymbol{x})} \sum_{i=t-k}^{t+k} \omega_{t-i}(\boldsymbol{y}_{\leq i}, \boldsymbol{x}) \log p_\theta(y_t \mid \boldsymbol{y}_{\leq i}, \boldsymbol{x})\right) \tag{5}$$

The weight function $\omega_{t-i}(\boldsymbol{y}_{\leq i}, \boldsymbol{x}) = \lambda_{t-i} \cdot c(\boldsymbol{y}_{\leq i} \mid \boldsymbol{x})$ is determined by the relative distance and direction of the dependency, as well as the confidence of the generated context $\boldsymbol{y}_{\leq i}$. Here, the factor $\lambda_{t-i} \in [0, 1]$ only depends on the relative position of the target token, decaying for longer dependencies. Using order-agnostic modeling, we calculate the confidence score $c$ by gathering the predicted probabilities based on different dependencies, which we obtain in the verification stage. Generally, backward reconstruction and next-token prediction based on iteratively refined content will be associated with higher weights. See Section 4.3 for a detailed comparison among different dependencies. In practice, some of the distributions in Eq. 5 may not be available for all tokens at each step. We calculate the ensemble utilizing available dependencies within the context window.

---

**Algorithm 1** Sliding Blockwise Order-Agnostic Decoding

---

1: **Input:** Order-agnostic generator $\pi_\theta$ and verifiers $v_\theta$ and $v_\theta^{\text{CD}}$ based on OA-LLM $p_\theta$, prompt $\boldsymbol{x}$, decoding context window size $k$, decoding block size $b$, maximum output sequence length $T$.
2: Initialize $t \leftarrow 0$, $\boldsymbol{y} \leftarrow \emptyset$.           ▷ Initialize the current length of the output sequence
3: Initialize $t_s \leftarrow 1$, $t_e \leftarrow \min(k, b)$. ▷ Initialize the start and end indices of the block to predict and refine
4: **while** $t_s < T$ **do**
5:   Construct $\mathcal{Y}_{t_s:t_e} \leftarrow \left\{ \{\tilde{y}_i\}_{i=t_s}^{t_e}, \tilde{y}_i \sim \pi_\theta(y_i \mid \boldsymbol{y}, \boldsymbol{x}) \right\}$.  ▷ Collect candidates through tree construction
6:   Select $\boldsymbol{y}_{t_s:t_e} \leftarrow \arg\max_{\tilde{\boldsymbol{y}}_{t_s:t_e} \sim \mathcal{Y}_{t_s:t_e}} \frac{1}{t_e - t_s + 1} \sum_{i=t_s}^{t_e} \left( v_\theta(\tilde{y}_i \mid \boldsymbol{y}, \boldsymbol{x}) + v_\theta^{\text{CD}}(\tilde{y}_i \mid \boldsymbol{y}, \boldsymbol{x}) \right)$.   ▷ Verify
7:   Update $\boldsymbol{y} \leftarrow \text{concat}(\boldsymbol{y}_{<t_s}, \boldsymbol{y}_{t_s:t_e})$.
8:   Set $t \leftarrow t_e$.
9:   **for** $i = t_s$ **to** $t_e$ **do**
10:     Sample $r \sim U[0,1]$ from a uniform distribution
11:     **if** $r < c(y_i \mid \boldsymbol{y}, \boldsymbol{x})$ **then**
12:       Set $t_s \leftarrow t_s + 1$.         ▷ Slide the decoding block based on rejection sampling
13:       **if** $y_i ==$ [EOS] **then**
14:         Exit while loop.
15:       **end if**
16:     **else**
17:       Exit for loop.
18:     **end if**
19:   **end for**
20:   Set $t_e \leftarrow \min(t_s + b - 1, t + k)$.
21: **end while**
22: **Output:** $\boldsymbol{y}$

---

**Verification.** Following Cai et al. (2024), we employ tree attention[3] to select from multiple candidates sampled from the ensemble distribution $\pi_\theta$. Each candidate is a combination of tokens used to fill the sliding block. Unlike previous works that only adopt the original next-token probability for verification, we also incorporate the backward reconstruction probabilities to leverage the refinement ability of COrAL. The verification score can thereby be formulated as follows:

$$v_\theta(y_t) = \frac{1}{\sum_{i=t-1}^{t+k} \lambda_{t-i}} \sum_{i=t-1}^{t+k} \lambda_{t-i} \log p_\theta(y_t \mid \boldsymbol{y}_{\leq i}, \boldsymbol{x}) \tag{6}$$

Here, we only consider the next-token and backward predictions for the verification score calculation. This scheme can be further enhanced by introducing a contrastive objective (Li et al., 2023) that penalizes the possible failure cases in forward multi-token prediction:

$$v_\theta^{\text{CD}}(y_t) = \max\left(0, \log p_\theta(y_t \mid \boldsymbol{y}_{\leq t-1}, \boldsymbol{x}) - \frac{1}{\sum_{i=t-k}^{t-2} \lambda'_{t-i}} \lambda'_{t-i} \log p_\theta(y_t \mid \boldsymbol{y}_{\leq i}, \boldsymbol{x})\right) \tag{7}$$

where $\lambda'_{t-i} = 1/\lambda_{t-i}$ to apply a higher penalty to predictions based on longer dependencies. Combining $v_\theta$ with $v_\theta^{\text{CD}}$, we keep the candidate of the highest average score. We allow several refinement iterations for each position within a sliding block to enhance the generation quality. Specifically, we propose an ensemble rejection sampling scheme to determine the sliding step size through majority voting across multiple dependencies, where we accept each token with the probability:

$$c(y_t \mid \boldsymbol{y}_{\leq t+k}, \boldsymbol{x}) = \frac{1}{k+2} \sum_{i=t-1}^{t+k} \mathbb{1}_{p_\theta(y_t \mid \boldsymbol{y}_{\leq i}, \boldsymbol{x}) > \min\left(\epsilon, \sqrt{\epsilon} \exp\left(-H\left(p_\theta(\cdot \mid \boldsymbol{y}_{\leq i}, \boldsymbol{x})\right)\right)\right)} \tag{8}$$

where $H(\cdot)$ is the entropy and $\epsilon$ is a fixed threshold to reject low-probability predictions. This acceptance scheme is inspired by truncation sampling (Hewitt et al., 2022; Cai et al., 2024) to choose candidates that are more likely to be sampled from the reference distributions. The sliding step size for each step is set to the length of the longest accepted prefix of the current block. We detail the sliding decoding procedure in Algorithm 1.

---

[3]To balance exploitation and exploration in tree construction, we select nodes according to the estimated accuracy of each token. Detailed considerations of candidate selection can be found in Appendix C.

Table 1: Result comparison of performance (accuracy %), speed (accepted tokens per second), and cost (seconds per sample) on arithmetic reasoning tasks. We compare against the conventional autoregressive greedy decoding approach as our next-token prediction baseline (NT). "verifier" and "multi-forward" represent the verification stage and multiple forward token prediction in inference.

| Approach | GSM8K | | | | MATH | | | |
|---|---|---|---|---|---|---|---|---|
| | Accu. | Speed | Speedup | Cost | Accu. | Speed | Speedup | Cost |
| NT | 74.1 | 39.7 | 1.0× | 3.67 | 21.8 | 38.7 | 1.0× | 5.41 |
| SC@4 | 76.2 | 37.8 | – | 15.5 | 23.0 | 38.0 | – | 16.6 |
| Ours | 75.3↑$_{1.2}$ | 43.4 | 1.1× | 3.35 | 22.7↑$_{0.9}$ | 44.4 | 1.1× | 4.82 |
| Ours w/o verifier | 72.4↓$_{1.7}$ | 156.8 | **3.9×** | 0.96 | 20.0↓$_{1.8}$ | 139.7 | **3.6×** | 1.47 |
| Ours w/o multi-forward | **78.7**↑$_{4.6}$ | 14.9 | – | 9.81 | **24.3**↑$_{2.5}$ | 11.5 | – | 18.2 |

## 4 EXPERIMENTS

In this section, we demonstrate the efficiency and breadth of COrAL regarding the quality–speed trade-offs across arithmetic, logical reasoning, and code generation.

**Datasets.** For arithmetic reasoning, we train COrAL on MetaMathQA (395K) (Yu et al., 2024) and evaluate it using GSM8K (Cobbe et al., 2021) on grade school math word problems and MATH (Hendrycks et al., 2021) of challenging competition mathematics problems. For logical reasoning, we filter LogiCoT (Liu et al., 2023b) with deduplication and reformulation, obtaining 313K training samples. We assess logical reasoning performance with multiple-choice reading comprehension tasks that test interpretation and decision-making skills: LogiQA (Liu et al., 2023a), based on the Chinese Civil Service Examination, and ReClor (Yu et al., 2020), sourced from Law School Admission Council exams. For code generation, we train on Magicoder-Eval-Instruct-110K (Wei et al., 2023) and evaluate using programming tasks from HumanEval (Chen et al., 2021).

**Experimental Protocol.** To address the discrepancy between the next-token-based pre-trained model and the target order-agnostic model, we adopt a two-stage training strategy (Kumar et al., 2022) to progressively enhance order-agnostic modeling. We begin with a domain-specific supervised fine-tuned (SFT) model. In the first stage, we perform order-agnostic training exclusively on the last target-aware layer, while freezing the other layers to preserve the output quality. In the second stage, we train the entire model by focusing on the previously frozen layers first and then unlocking the last layer to train together. We use Mistral-7B-v0.3 and DeepSeek-Coder-6.7B-base as the base models for reasoning and code generation tasks, respectively. During inference, we explore the effect of the verification stage and ablate the values of decoding context window size and block size. Given the order-agnostic training tax resulting from the discrepancy between pretraining and fine-tuning objectives, we use next-token prediction with the same model as the baseline to ensure a fair comparison. We detail our hyperparameter settings in Section 4.2 and Appendix D.

### 4.1 MAIN RESULTS

We compare our order-agnostic decoding approach (Section 3) with its next-token counterparts across three tasks. We also show the quality–speed trade-offs in by ablating the decoding settings.

**Arithmetic Reasoning.** As shown in Table 1, COrAL enhances the effectiveness and efficiency through different mechanisms in order-agnostic generation. Using both verification and multiple forward token prediction in decoding, COrAL surpasses the corresponding next-token baseline with comparable inference-time cost. Furthermore, by trading inference speed with iterative generation and verification through backward refinement, we observe a substantial improvement in accuracy from 74.1% to 78.7 and 21.8% to 24.3%, on GSM8K and MATH, respectively. When skipping the verification stage for quality control, our approach significantly speeds up the decoding process up to 3.9×. This demonstrates the flexibility of COrAL in enhancing both the generation quality and inference speed in mathematical reasoning.

**Logical Reasoning.** Table 2 compares the performance and generation speed of model outputs under different decoding settings on logical reasoning tasks. Similarly, COrAL improves the reasoning

Table 2: Result comparison of performance and speed on logical reasoning tasks.

| Approach | LogiQA | | | ReClor | | |
|---|---|---|---|---|---|---|
| | Accu. | Speed | Speedup | Accu. | Speed | Speedup |
| NT | 55.1 | 33.6 | 1.0× | 63.2 | 33.2 | 1.0× |
| Ours | $58.2\uparrow_{3.1}$ | 62.1 | 1.8× | $62.7\downarrow_{0.5}$ | 38.2 | 1.2× |
| Ours $_{\text{w/o verifier}}$ | $55.7\uparrow_{0.6}$ | 99.1 | **2.9×** | $61.6\downarrow_{1.6}$ | 72.0 | **2.2×** |
| Ours $_{\text{w/o multi-forward}}$ | $\mathbf{59.1}\uparrow_{4.0}$ | 8.9 | – | $\mathbf{64.7}\uparrow_{1.5}$ | 11.3 | – |

Table 3: Result comparison of pass rates and speed on code generation.

| Approach | HumanEval | | |
|---|---|---|---|
| | Pass@1 | Speed | Speedup |
| NT | **64.6** | 42.2 | 1.0× |
| Ours | $13.0\downarrow_{51.6}$ | 45.8 | 1.1× |
| Ours $_{\text{w/o verifier}}$ | $6.5\downarrow_{58.1}$ | 119.0 | **2.8×** |
| Ours $_{\text{w/o multi-forward}}$ | $61.6\downarrow_{3.0}$ | 28.8 | – |

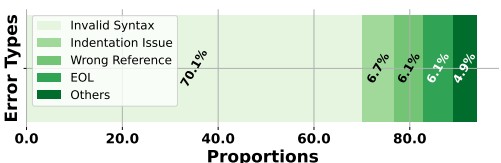

Figure 4: Meso-analysis of error cases in code generation (Ours $_{\text{w/o verifier}}$) on HumanEval. The primary failure cases come from syntax errors.

performance by augmenting next-token prediction exclusively with backward refinement. However, we observe a discrepancy in the performance improvements on LogiQA and ReClor with absolute increases of 4.0% and 1.5% in corresponding accuracies. We attribute this gap to the imbalanced proportions of the two tasks in our SFT data from LogiCoT (Liu et al., 2023b). This also implies the importance of high-quality data selection to boost the effect of order-agnostic training to model different dependencies related to the target tasks.

**Code Generation.** Results on code generation, however, show an opposite effect of order-agnostic modeling on performance. In Table 3, we observe substantial performance drops across different decoding settings using COrAL. For example, without verification, the pass rate on HumanEval decreases to 6.5% from 64.6% of next-token prediction. This gap remains to be large when applying verification for quality control. Error analysis in Figure 4 indicates that the major cause of this drop comes from the erroneous syntax, where the primary error type, *Invalid Syntax*, accounts for 70.1% of all samples. To mitigate this issue, we can turn off the mechanism of forward multi-token prediction and increase the threshold $\epsilon$ in Eq. 8 to reject tokens with low confidence scores. For example, with $\epsilon = 0.5$, COrAL achieves a comparable pass rate of 61.6% compared to 64.6% of the baseline. The absolute decrease of 3.0% indicates the deficiency of COrAL in producing incoherent content, showing the importance of specific designs for tasks requiring strict textual formats.

### 4.2 Ablation Studies

In this section, we analyze the core designs of COrAL to enable efficient iterative refinement. We probe the effect of different training and decoding hyperparameters.

**Backward Refinement Improves Generation Quality.** Figure 1 shows how the performance and inference cost scale with iterative refinement. Note that even without backward dependencies in prediction, COrAL can still perform backward refinement using the next-token prediction. In this case, we examine the effect of backward dependencies with different context window sizes. Notably, the performance of iterative refinement scales faster than the inference cost as the iteration time increases. Furthermore, leveraging backward dependencies, COrAL reaches a higher plateau of performance compared to refining with forward dependencies only. However, the fast saturation of performance improvement with larger refinement times indicates a relatively low upper bound of the enhancement brought by backward reconstruction. We extensively discuss this problem attributed to the discrepancy between pre-training and fine-tuning objectives in Appendix E.

**Quality–Speed Trade-off in Inference.** In Section 4.1, we demonstrate the quality–speed trade-off by ablating the employment of verification and forward multi-token prediction. We now provide a detailed analysis of the decoding hyperparameters to show this trade-off. We consider the block size $b$ and the forward context window size $k$, two variables closely related to the inference speed and quality. For block size, we probe its effect in reducing inference time in the verification-free case

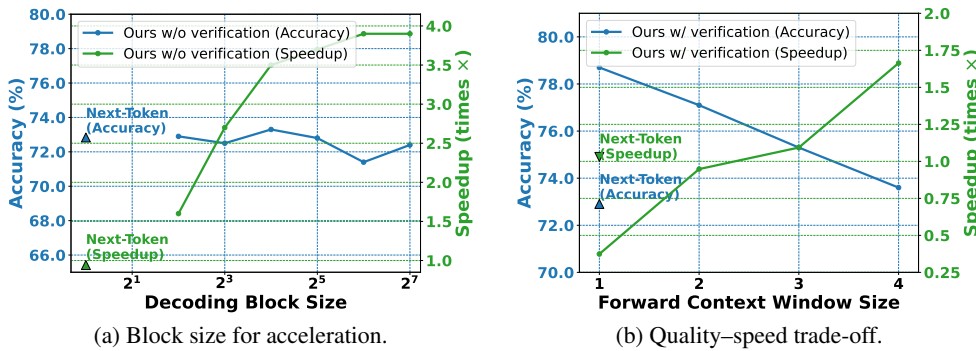

(a) Block size for acceleration.     (b) Quality–speed trade-off.

Figure 5: Quality–speed trade-offs on GSM8K. Generally, COrAL accelerates inference with larger decoding block size $b$ and forward context window size $k$.

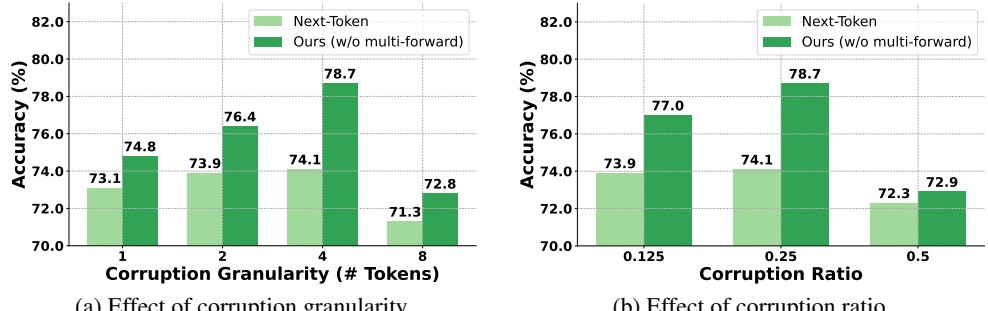

(a) Effect of corruption granularity.     (b) Effect of corruption ratio.

Figure 6: Ablation on the effects of corruption granularity and ratios in training for backward reconstruction. We probe the variation in model improvements from backward dependencies.

to maximize the speedup rate. Figure 5a shows that we can push the speedup boundary toward the corresponding upper bound of $k$ with large $b$. For example, given $k=4$, we approach the maximum speedup rate $4\times$ with large block sizes such as $b=64$ and $128$. Notably, leveraging the backward refinement capability of COrAL, this block size-driven acceleration process retains the generation quality at the same level, illustrating the balance of efficiency and effectiveness of our acceleration mechanism. For forward context window size, we adopt the two-stage prediction–verification setting to explore the quality improvement boundary of the iterative refinement mechanism. Figure 5b shows trends of performance drop and inference speedup when increasing $k$. We explain this trade-off as a reflection of the decreasing precision in predicting future tokens of longer dependencies.

**Learning from Corruption Enhances Refinement Capability.** One core design in COrAL is the denoising process to enable iterative refinement, where the corruption strategy is crucial for controlling data quality and model performance. In Figure 6, we analyze the variation in performance improvements from backward refinement (w/ multi-forward) when applying corruption with different granularity or ratios. Given a backward context window size $k = 8$, we observe a more significant improvement when applying corruption on longer pieces of text. For example, COrAL achieves an absolute increase of $4.6\%$ with granularity 4, compared to $1.7\%$ under token-level corruption. However, as the corrupted context gets longer, the model's capability to learn from mistakes may also degrade. One possible reason for this performance drop is the difficulty and inconsistency in simultaneous multi-token regeneration, as reconstructing more tokens brings higher uncertainty and noise. This indicates the importance of using a reasonable corruption granularity to obtain data of good quality and maintain training stability. Likewise, we see a similar trend when the corruption ratio varies. Specifically, a high corruption ratio such as $0.5$ can damage the semantic meaning of the context, leading to a performance drop in both our and baseline approaches. Nevertheless, we can still benefit when increasing the corruption ratio within a reasonably lower range, such as $0.125$ to $0.25$, to enhance the reconstruction process.

### 4.3 Further Analysis

We analyze COrAL's capability to model different dependencies, and the potential computation overhead from order-agnostic modeling.

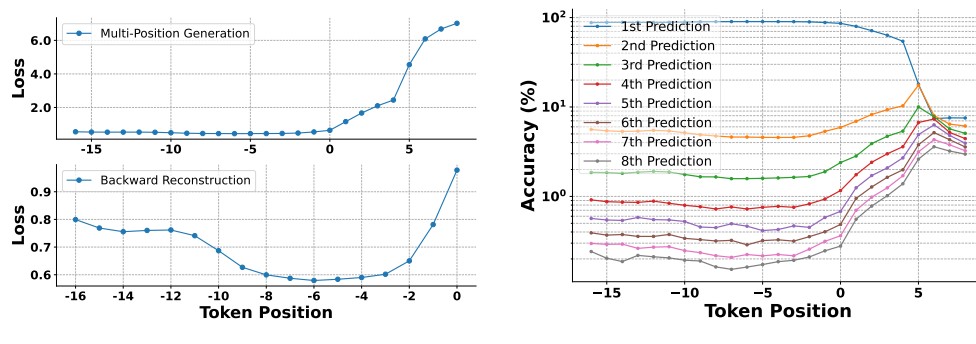

(a) Token-wise loss.   (b) Token-wise accuracy of top-8 predictions.

Figure 7: Token-wise losses and accuracies corresponding to different dependencies.

**How does COrAL model order-agnostic dependencies?**   We compare the model capabilities across different positions using token-wise losses and accuracies in Figure 7. Generally, COrAL performs better on backward reconstruction than forward prediction, as shown in the lower losses and higher accuracies on backward dependencies. Notably, we see better generalizability of backward reconstruction. For example, given the backward context window size $k = 8$ and forward context window size $k = 4$, we find that the loss and accuracy of backward reconstruction with dependencies longer than the training context window size, such as positions $|-9| > |-8|$, are also at the same level as other backward dependencies. Differently, we observe a dramatic increase in loss and a drop in accuracy from positions $4$ to $5$ on longer dependencies in forward prediction. This explains how backward refinement benefits from more information in sequence-level generation to improve performance. We observe decreased performance for forward prediction as the dependency gets longer, especially when it exceeds the forward context window size in training. However, we can mitigate this issue by aggregating multiple predictions for each position. As shown in Figure 7b, while forward positions with longer dependencies obtain lower accuracies on tghe first prediction, the accumulated accuracies of their non-first predictions are generally higher than those from other dependencies. This illustrates how COrAL can benefit from the tree construction and verification stage in decoding (Section 3) by considering multiple candidates for each position.

**Computation Overhead.**   One concern regarding order-agnostic modeling is the potential computation overhead to accommodate more dependencies in the context windows. As target-aware RoPE is only applied on the last layer, this overhead scales relatively slower as we increase the number of positions to predict. For example, with forward and backward context window sizes each set as $k = 4$, each forward pass of COrAL costs $5.48$ TFLOPS, compared with $2.81$ TFLOPS of next-token prediction. In other words, COrAL predicts $8\times$ number of tokens with less than $2\times$ overhead in computational cost. This indicates the efficiency of COrAL in leveraging available computation resources to accelerate and enhance inference. Furthermore, we can adjust the forward and backward context window sizes to determine the number of tokens to predict in parallel, demonstrating the flexibility and generalizability of COrAL with target-aware RoPE.

## 5   CONCLUSION AND FUTURE WORK

By unifying denoising with context-wise order-agnostic language modeling and introducing target-aware positional encoding, COrAL incorporates iterative refinement directly into the language generation process while keeping inference costs low. This approach offers a promising direction for developing more efficient and capable large language models by effectively capturing local dependencies within context windows and reducing inference latency.

The effectiveness and efficiency of COrAL underscores the promise of order-agnostic strategies as a generalized architecture to facilitate generative language modeling and text generation. Specifically, it suggests new opportunities to unify: (**i**) the sequence modeling and varying-length generation abilities of autoregressive modeling, (**ii**) the multi-dependency modeling and multi-token prediction mechanisms in order-agnostic modeling, and (**iii**) the efficient way of iterative refinement in denoising techniques. We hope our work will motivate future research to explore order-agnostic modeling and denoising in various tasks and other domains beyond sequence modeling.

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

LIMITATIONS

This work proposes an approach to integrate iterative denoising with order-agnostic language modeling to enhance both the effectiveness and efficiency of LLM inference. While it offers a promising paradigm for mitigating issues related to monotonic dependencies and inference latency in conventional autoregressive models, several directions remain for further exploration, including designing corruption and decoding strategies to tailor the model to specific tasks, optimizing the training process to overcome the order-agnostic training tax, and probing the generalizability and scalability of COrAL across different context sizes, model scales, and tasks.

Specifically, order-agnostic language modeling can struggle with tasks that demand specific output formats or syntax due to inconsistencies in the multi-token predictions. This indicates the importance of a task-specific design of the acceptance scheme in order-agnostic decoding. For instance, the performance of the verification policy in Eq. 6 may vary by language and domain. Additionally, applying semantic-aware weights to different dependencies could further enhance task-specific features in the generated outputs. Future work can further explore the potential of incorporating different evaluation heuristics to guide the inference process.

Furthermore, incorporating corrupted data may introduce discrepancies between training- and inference-time objectives. For example, our experiments explore rule-based context-wise corruption strategies to create noisy data. Future work could focus on diversifying the types of corruption and scaling the difficulty level and proportion to better understand their impacts on model capabilities.

Finally, due to the computation constraint, we explore the model capabilities in order-agnostic modeling with fixed context window sizes during the SFT stage only. Future work may investigate the effect of scaling context window sizes in both forward and backward directions. Moreover, increasing the context window sizes may exacerbate the discrepancy between autoregressive pre-training and order-agnostic fine-tuning. We thus anticipate future work to extend COrAL to the pre-training stage to further enhance model capabilities.

POTENTIAL BROADER IMPACT

Compared to conventional autoregressive modeling, COrAL leverages multi-token prediction and reconstruction to backtrack and iteratively refine past generations. This strategy mirrors the human decision-making process in real-world task completion. We anticipate COrAL to inspire the community to design more efficient and effective frameworks to enhance interpretability and alignment with the reasoning and planning process of humans.

## A  CONCEPTUAL COMPARISON AMONG MODEL ARCHITECTURES

We consider the properties an ideal architecture should have as follows:

- **VL**: varying-length generation
- **BT**: backtrack / look-ahead
- **MV**: multi-variable generation
- **MD**: multi-dependency (inter-sample connection) modeling
- **FS**: fitting feasibility
- **EF**: inference efficiency
- **IT**: mechanism of iterative refinement

Table 4: Conceptual comparison regarding desired features across different architectures.

| Architectures | VL | BT | MV | MD | FS | EF | IT |
|---|---|---|---|---|---|---|---|
| Next-Token AR (Uria et al., 2016) | ✓ | ✗ | ✗ | ✗ | ✓ | ✗ | ✗ |
| Permutation-Based AR (Uria et al., 2014) | ✗ | ✓ | ✓ | ✓ | ✗ | ✓ | ✗ |
| NAR (Gu et al., 2018) | ✗ | ✓ | ✓ | ✓ | ✓ | ✓ | ✓ |
| Diffusion (Ho et al., 2020) | ✗ | ✓ | ✓ | ✓ | ✓ | ✗ | ✓ |
| Consistency Model (Song et al., 2023) | ✗ | ✓ | ✓ | ✓ | ✓ | ✓ | ✓ |
| COrAL (Ours) | ✓ | ✓ | ✓ | ✓ | ✓ | ✓ | ✓ |

## B   FURTHER RELATED WORK

**Order-Agnostic Language Modeling.**   Order-agnostic architectures have been explored to overcome the limitations of sequential generation in autoregressive models. Uria et al. (2014) propose permutation-based autoregressive models to learn different data orderings for density estimation. In language modeling, Yang et al. (2019) further explore the idea of order-agnostic autoregressive modeling as a generalized pretraining method. Welleck et al. (2019) explore the possibility of non-monotonic text generation in a tree-structure manner and achieve competitive performance with the conventional left-to-right sequential generation. To avoid the high latency in autoregressive decoding, Gu et al. (2018) introduce non-autoregressive machine translation by breaking the sequentially causal dependency across time into conditionally independent per-step distributions with latent variables as intermediate steps. Lee et al. (2018) adopt iterative refinement to interpret the latent variable model, inspired by the design of denoising autoencoders (Alain & Bengio, 2014). Follow-up works on non-autoregressive machine translation show promising performance of the iterative refinement process of mask-predict (Ghazvininejad et al., 2019; Kasai et al., 2020). Our work explores the potential of unifying the strengths of order-agnostic modeling and denoising to advance sequential modeling in LLMs, demonstrating an efficient way to conduct iterative refinement internally.

**LLM Self-Refinement.**   Self-refinement in LLMs focuses on various feedback mechanisms to improve the model performance dynamically. Existing works utilize the feedback mainly in two directions. The first one relates to prompting-based frameworks such as instance-level refinement (Madaan et al., 2023), step-level guided search (Yao et al., 2023; Xie et al., 2023), and principle-driven reasoning (Zheng et al., 2024). Another line of work adapts the feedback as training signals to further enhance the performance of LLMs, including rationale-augmented refinement (Zelikman et al., 2022), hindsight-driven alignment (Zhang et al., 2023; 2024c), and search-enhanced preference learning (Xie et al., 2024). Unlike existing works relying on the AR foundation in conventional LLMs, we leverage the order-agnostic modeling ability of COrAL to conduct the iterative refinement internally while foregoing the computation overhead in AR-LLMs to maintain efficiency.

**Parallel Decoding.**   Parallel decoding methods aim to accelerate LLM inference by generating multiple tokens simultaneously rather than sequentially. Non-autoregressive models (Gu et al., 2018) and blockwise decoding approaches (Stern et al., 2018; Monea et al., 2023; Cai et al., 2024) have enabled faster generation but often struggle with output inconsistencies. Speculative decoding techniques (Leviathan et al., 2023; Chen et al., 2023; Miao et al., 2024) adopts a faster draft model to speedup inference while struggling with the deficiency in scalability. Self-speculative decoding (Zhang et al., 2024a) uses the same model for drafting by selectively skipping certain intermediate layers. Look-ahead (Santilli et al., 2023) and Jacobi (Fu et al., 2024) decoding, on the other hand, directly utilize the AR LLMs to enhance performance iteratively. Consistency LLMs (Kou et al., 2024) further reduces this iteration time drawing inspiration from consistency models (Song et al., 2023; Song & Dhariwal, 2024). In this work, we realize parallel decoding leveraging the multi-token generation ability of COrAL. Instead of decoding toward the forward direction only, we support backward refinement simultaneously to enhance the generation quality further.

**Iterative Refinement.**   Prompt engineering approaches (Madaan et al., 2023; Shinn et al., 2023) exploit incorrect attempts in historical data to improve the performance of a frozen LLM. In contrast, our method enables the model to directly correct generated mistakes via backward refinement. Verifier-based methods (Cobbe et al., 2021; Lightman et al., 2024) train separate models to re-rank outputs. These strategies are orthogonal to our method, which could further enhance COrAL by providing stronger verification mechanisms. An et al. (2024) demonstrate that the mistake reasoning data can be directly utilized through a standard fine-tuning approach. However, this approach relies on AR-LLMs and sequential prediction, whereas COrAL introduces a fundamentally new paradigm by enabling mistake correction through backward dependencies.

**Scheduled Sampling.**   Scheduled sampling (Bengio et al., 2015) aims to mitigate the discrepancy between training and inference, it gradually transitions from teacher-forcing to self-generated inputs using curriculum learning. In contrast, COrAL decomposes the order-agnostic training into two separate objectives: forward prediction with ground-truth input and backward reconstruction with corrupted input. Inspired by scheduled sampling, future iterations of COrAL could explore cur-

riculum strategies to gradually increase corrupted input ratios, enhancing robustness and stability. Furthermore, scheduled sampling is designed for sequential decoding at inference, while COrAL employs blockwise order-agnostic decoding, enabling multi-token forward prediction for speedup and backward refinement for quality improvement.

## C  CANDIDATE TREE CONSTRUCTION IN ORDER-AGNOSTIC DECODING

Our specific design of tree construction aims to explore promising combinations of multi-position predictions with a fixed budget for the number of total nodes in the tree. Unlike selecting promising nodes based on the accuracies of the top predictions of different heads in Cai et al. (2024), we forego the need of a validation set for accuracy calculation by leveraging the model confidence of each prediction with a dedicated scaling factor. Let $p_t^{(i)}$ denote the model-predicted probability of the $i$-th top candidate for the $t$-th token. For a candidate sequence composed by the top $[i_{t_s}, i_{t_s+1}, \cdots, i_{t_e}]$ predictions of tokens at different positions, we estimate its accuracy as:

$$\prod_{j=t_s}^{t_e} \left( p_j^{(i_j)}/\gamma_j \right) \tag{9}$$

where $\gamma_i$ is a scaling factor to up weight the predictions based on nonconsecutive forward dependencies. As shown in Figure 7, this process benefits from the fact that COrAL obtains higher accuracies on non-first predictions on such dependencies. Empirically, we set these factors to be 1.1, 1.2, 1.3 for the second, the third, and the fourth tokens to predict, respectively.

Following Eq. 9, we construct the tree in a greedy manner, adding the node with the highest confidence to the tree one by one. This process considers the token-wise confidence as the expected contribution of each prediction to the tree. We repeat the node-adding process until the total number of nodes reaches the desired number to accommodate the maximum sequence length the model can deal with.

## D  HYPERPARAMETER SETTING

**Training.**  For order-agnostic training, we train for 3 epochs at each stage with a batch size of 128 on all tasks. We fix the context window size in training as 4 and 8 for forward and backward dependencies. At different training stages, we recommend employing different learning rates. We set the learning rate as 5e−6 and 1e−4 in reasoning and code generation, respectively, for the last-layer tuning stage. We increase the learning rates at the second stage to be 1e−6 and 2e−5 for corresponding tasks following the general SFT settings.

We corrupt the training data with granularity 4 and ratio 0.25 across all tasks for backward reconstruction. As discussed in Section 4.2, we ablate the granularity and ratios on mathematical reasoning data to study their respective effects on enhancing model's refinement abilities. Note that as the training context window sizes are fixed as 4 and 8 for forward and backward dependencies, the optimal corruption hyperparameters may vary as we scale the context window sizes. Due to the computation constraint, we leave it to future work to explore the combinations of different granularities and corruption strategies.

**Decoding.**  For order-agnostic decoding, we suggest adopting different context window sizes and block sizes to balance the quality and inference speed in different tasks. We report the experiment results (Section 4.1) under the same context window and block sizes across the three tasks, where the forward and backward context window and block sizes are 4, 8, and 64, respectively. For verification, we set $\epsilon = 0.2$ and 0.5 for reasoning and code generation. We implemented our order-agnostic decoding and corresponding next-token baseline without KV-Cache (Pope et al., 2023). During decoding, we set the batch size 1 and conduct inference on a single GPU.

**Computation.**  For reasoning tasks with maximum sequence length 512, all training experiments were done on single-node eight 40GB A100s. For code generation task with maximum sequence length 2048, we conduct training and inference on single-node four 80GB H100s.

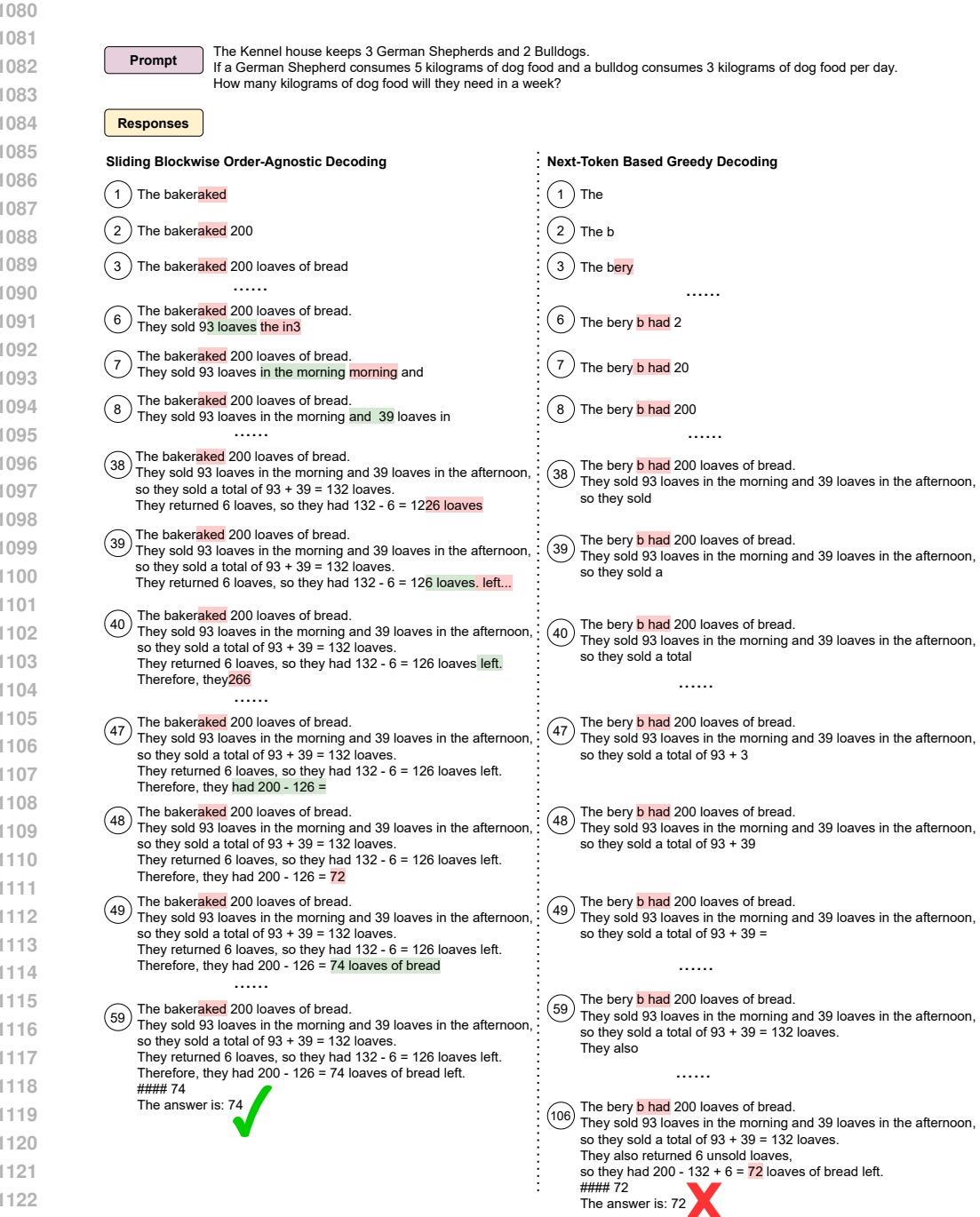

Figure 8: Qualitative result comparison on GSM8K.

# E   EXTENDED DISCUSSION

In this section, we extensively discuss the training protocol we design to endow AR-LLMs with order-agnostic ability without pretraining. Lastly, we illustrate how COrAL efficiently corrects mistakes in previous generations in qualitative analysis.

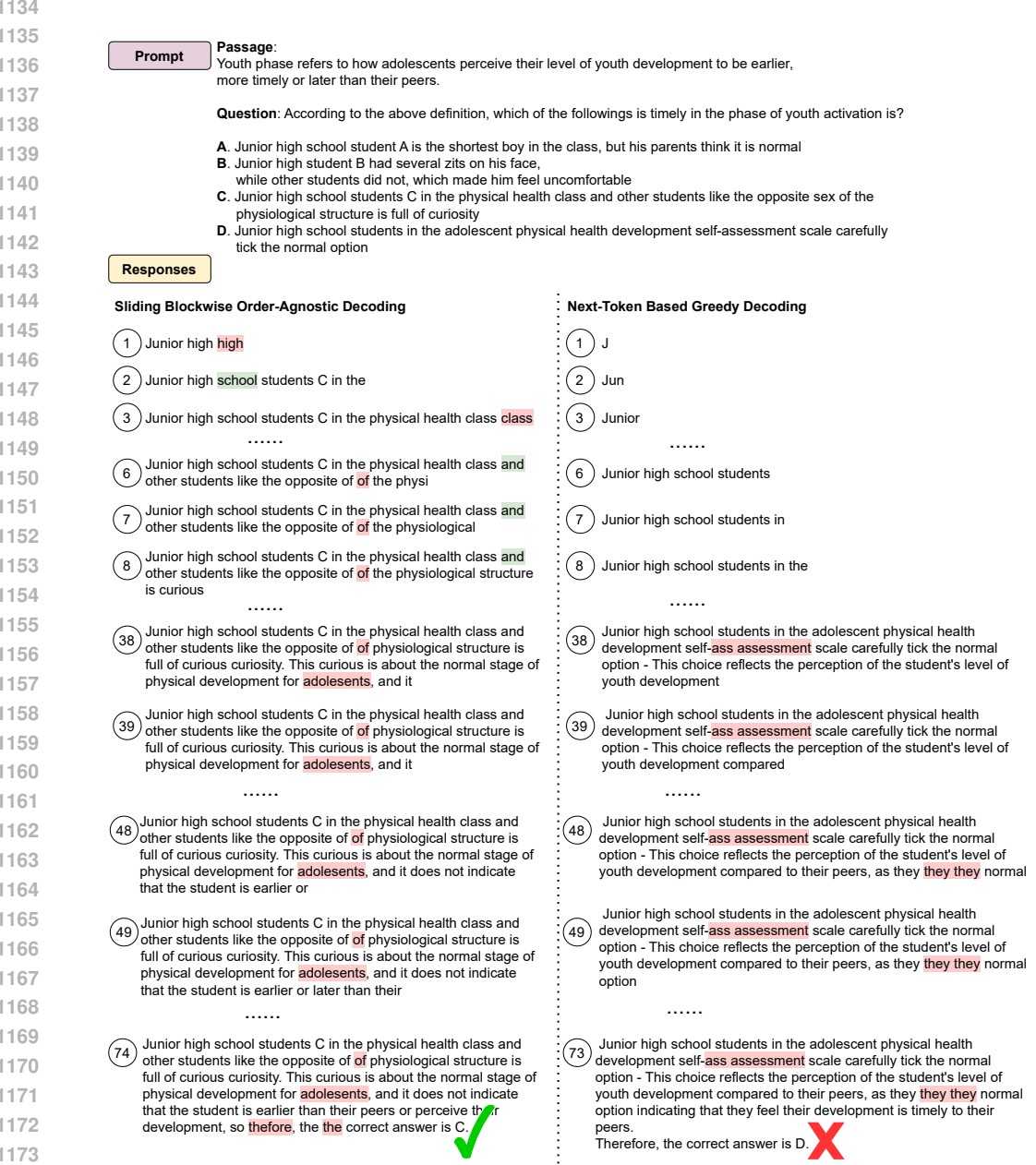

Figure 9: Qualitative result comparison on LogiQA.

**Effect of Two-Stage Training.** As discussed in Section 4.2, a high corruption ratio can cause a collapse in model performance as the noisy data contains corrupted information in a format that the model has not seen in pretraining. Furthermore, we are also faced with the order-agnostic training tax to endow an AR-based LLM with denoising and multi-token prediction abilities. In this section, we elaborate on the two-stage training we designed to mitigate this issue. Following Cai et al. (2024), we first tune the last layer where we apply target-aware RoPE. This adapts the previous parameterization on next-token prediction to target-aware multi-position prediction. Due to the discrepancy of training objectives in pretraining and fine-tuning, full fine-tuning is still essential to ensure better performance on multi-token prediction. To stabilize the training process, we then freeze the last layer and gradually unlock it through the second training stage of full fine-tuning.

Empirically, we find this strategy effective for stabilizing the autoregressive loss changes in forward prediction. However, we observe an order-agnostic training tax where the next-token prediction performance drops from $77.0\%$ to $76.5\%$ and then $74.1\%$ after the first and second stages, respectively. This performance degradation possibly comes from two aspects: the difference in training objectives and the incorporation of corrupted data in fine-tuning. We leave it to future work to further explore the effect of applying our order-agnostic framework to the pretraining stage.

**Qualitative Analysis.** Our qualitative analysis on GSM8K and LogiQA showcases how COrAL corrects previously generated mistakes through the iterative internal process. In Figure 8, COrAL obtained a wrong calculated result 72 at the 48-th step. However, the backward refinement mechanism enables it to backtrack and refine the result to the correct number, 74, as shown at the 49-th step. In contrast, the next-token baseline cannot correct the erroneous 72, leading to the wrong final result. On the other hand, we observe the incoherence in COrAL's generation where COrAL can fail in correcting the mistakes when it happens to skip some positions during generation. For example, at the 1-st step, COrAL outputs "bakeraked" instead of "baker baked". This error incurs a chain reaction where the subsequent outputs all omit the correct token " b" right after "baker", indicating the need for further enhancement on the generation fluency of order-agnostic methods.

On LogiQA, interestingly, we observe a higher frequency of the inconsistencies in COrAL's generation. As discussed in Section 4.1, we attribute this scenario to the relatively low proportion of LogiQA-related training data in LogiCoT, where there are only 5K samples out of the 313K data points. As shown in Figure 9, while the COrAL produces several grammatical errors in a generation, it still achieves the correct result. This indicates the advanced ability of COrAL to sematically escape from paths that may lead to dead ends through iterative refinement.

**Further Analysis on Computation Overhead.** As discussed in Section 4.3, the computation overhead in COrAL scales efficiently relative to the number of predicted positions, with target-aware RoPE applied only to the last layer. We now provide a detailed computation comparison between COrAL and the baseline approaches.

Table 5: Computation comparison across different decoding approaches on GSM8K.

| Approach | TFLOPS (per forward pass) | Accuracy (%) | Speed (tokens per second) | Speedup |
|---|---|---|---|---|
| Next-Token (NT) | 2.81 | 74.1 | 39.7 | 1.0× |
| Ours | 13.6 | 75.3 | 43.4 | 1.1× |
| Ours w/o verifier | 5.48 | 72.4 | 156.8 | 3.9× |
| Ours w/o multi-forward | 17.9 | 78.7 | 14.9 | − |

