# OpenReview forum: "COrAL: Order-Agnostic Language Modeling for Efficient Iterative Refinement"
_ICLR.cc/2025/Conference — Submitted to ICLR 2025_

### Official Review · Reviewer_LRiM · 2024-11-02

**Soundness:** 4
**Presentation:** 2
**Contribution:** 3
**Rating:** 6
**Confidence:** 3

**Summary:**

The authors propose a new decoding method, called CORAL, which can speed up the decoding process and maintain (or upraise) the performance of the model in some tasks. CORAL has 2 parts: prediction and verification. The experiment shows that the verification part can help the model to generate more accurate results. CORAL also designed a strategy named "multi-forward" to speed up the decoding process (although it may hurt the performance). The result shows that the CORAL is useful in math problems but is useless in the code generation task.

**Strengths:**

1. The topic of the paper is interesting, transformer-based model do have the problem of slow decoding speed.

2. It make a good balance between the speed and the performance.

**Weaknesses:**

1. The improvement of the CORAL is not generalizable enough. It only works well in the some math/logic problems but not in the code generation task.

2. Although the speed of the decoding process is improved, it needs to use more GPU memory (and "waste" some computation because of verification and multi-forward) to achieve this. So it is not friendly to equipment that most people use.

**Questions:**

In eq.8 entropy is always positive, so -H(x) is always negative and exp(-H(x)) is always less than 1. So min(a,a*exp(-H(x))) is always a*exp(-H(x)).

---

> ### Author Response · Authors · 2024-11-21
> **Response to Reviewer LRiM**
>
> We appreciate your constructive feedback!
>
> > The improvement of the CORAL is not generalizable enough
>
> We appreciate the reviewer’s concern about the generalizability of COrAL.
>
> We would like to highlight that COrAL achieves significant improvements in accuracy and inference speed on various reasoning tasks, spanning from arithmetic reasoning, including GSM8K ($+4.6$%) and MATH ($+2.5$%), to logical reasoning containing LogiQA ($+4.0$%) and ReClor ($+1.5$%). Our comprehensive experiments demonstrated the potential of order-agnostic language modeling to enhance reasoning through internal iterative refinement. On the other hand, our extended experiment on code generation also shows the limitations of COrAL in tasks that require strict syntactic coherence, providing a deeper understanding of the pros and cons of our proposed method.
>
> Due to the constraints in computation and data resources, we leave it to future work to explore a broader range of tasks (e.g., instruction following, writing, dialogue, infilling) to probe both the generalizability and specialty of COrAL. We have included a detailed discussion on this in the Limitations of our paper.
>
> > it needs to use more GPU memory (and "waste" some computation because of verification and multi-forward) to achieve this. So it is not friendly to equipment that most people use.
>
> Thanks for raising your concerns about the GPU memory and computational overhead in COrAL.
>
> As discussed in Section 4.3, this overhead scales efficiently relative to the number of predicted positions, with target-aware RoPE applied only to the last layer. Take results on GSM8K for example:
>
> | Approach | TFLOPS (per forward pass) | Accuracy (%) | Speed (tokens per second) | Speedup |
> | :- | :-: | :-: | :-: | :-: |
> | NT    |                                                $2.81$   | $74.1$ | $39.7$ | $1.0\times$ |
> | Ours |                                                $13.6$   |  $75.3$ | $43.4$ | $1.1\times$ |
> | Ours $_\textrm{w/o verifier}$ |           $5.48$  | $72.4$ | $156.8$ | $3.9\times$ |
> | Ours $_\textrm{w/o multi-forward}$ | $17.9$  | $78.7$ | $14.9$ | $-$ |
>
> With forward and backward context window sizes of $k = 4$, COrAL (w/o verifier) costs $5.48$ TFLOPS per forward pass compared to $2.81$ TFLOPS for next-token prediction. In other words, COrAL predicts $8\times$ number of tokens with less than $2\times$ overhead in computational cost. This indicates the efficiency of COrAL in leveraging available computation resources to accelerate and enhance inference.
>
> Moreover, users can adjust the decoding hyperparameters (e.g., context window and block sizes) to suit their device capabilities. For reference, our experiments used a forward and backward context window of $k=4$, a block size of $b=64$, and a maximum sequence length of $512$ on a 40 GB A100 GPU for reasoning tasks.
>
> > In eq.8 entropy is always positive, so -H(x) is always negative and exp(-H(x)) is always less than 1. So min(a,aexp(-H(x))) is always aexp(-H(x)).
>
> Thanks for pointing out the typo. The correct formulation should be $\min(\epsilon, \sqrt{\epsilon}\exp(-H(x))$ instead. We have corrected this typo in the updated manuscript.

---

> > ### Comment · Reviewer_LRiM · 2024-11-22
> >
> > Thanks for reviewer's reply. I will raise my confidence score.

---

> ### Author Response · Authors · 2024-11-25
> **Thanks for the update and valuable feedback**
>
> Dear Reviewer LRiM
>
> Thank you very much for the valuable feedback and the update on the confidence score. We are happy to know that you found the response satisfactory. We have included a detailed discussion about the generalizability of COrAL in the Limitations and a more thorough comparison of the computation overhead of different decoding approaches in Appendix E. We hope you might view this as sufficient reason to further raise your score.
>
> Best,
>
> Authors

---

> > ### Author Response · Authors · 2024-11-29
> >
> > Dear Reviewer LRiM,
> >
> > Thanks again for your valuable feedback and recognition of our contributions.
> >
> > With the extended discussion period, we would like to engage in further discussions and address any remaining questions or concerns!
> >
> > Best,
> >
> > Authors

---

### Official Review · Reviewer_oenX · 2024-11-03

**Soundness:** 3
**Presentation:** 3
**Contribution:** 3
**Rating:** 8
**Confidence:** 3

**Summary:**

The authors introduce a novel decoding strategy combining autoregressive modelling with ROBERTA-like order agnostic refinement. Given a partial sequence, they predict multiple tokens ahead, which they subsequently refine using ROBERTA-like denoising autoencoder. The authors see performance improvements on GSM8K and LogiQA and poor performance on code generation.

**Strengths:**

* the authors propose an interesting paradigm and show that it has promise for reducing computational cost and enhancing performance in certain settings
* the method is applicable to autoregressive pretrained language models and seems to improve their performance in certain settings
* the authors provide a quite extensive ablation study for their method
* the paper contains some beautiful figures such as figure (2) and (3). Even though Figure (2) is a little bit unclear to me. Why are there seemingly different offsets for the refinements and why is there not much visual seperation inbetween forward prediction and refinement?

**Weaknesses:**

* pseudo-code for Algorithm 1 is provided without walking through the pseudo-code
* in the experimental section the baselines are not described in enough detail, just AR. the proposed method requires finetuning, are the AR baselines also finetuned on the tasks?
* the by-far-best performance is achieved using the w\o multi-head prediction ablation, which is not the proposed method and thus weird. I assume this variant suffers from increased computational cost compared to the proposed method. It would be interesting to compare this ablation with a method from the related work that has a similar computational cost.
* comparison to refinement methods from the related work is missing
* a somewhat non-standard notation for expected values is used. their subscripts seem to be used much like in summations, but usually subscripts at an expected value are used to indicate over which distribution the expectation is taken: e.g., equation (1) and equation (3)

**Questions:**

It would be really interesting to check how much performance is lost by starting from a pretrained model as compared to full training a method employing coral from scratch. Do you think that some performance is left on the table because you start from a pretrained model?

In the main result part, to increase my rating I would like to see a comparison to other interative refinement methods that have a similar computational cost as the w/o multi-token prediction variant of the proposed method and also a more detailed description of the autoregressive baseline.

Suggestion: Maybe it would be a good idea to incorporate an application in which this method shines. E.g., by looking into domains that can benefit from the order-agnostic aspect such as protein language modelling.

---

> ### Author Response · Authors · 2024-11-21
> **Response to Reviewer oenX (1/2)**
>
> Thanks for your insightful and thorough comments!
>
> > Are the AR baselines also fine-tuned on the tasks?
>
> Yes, the AR baselines are based on the same order-agnostic model, fine-tuned on tasks with both forward prediction and backward refinement objectives (Section 2). To ensure fair comparison, we adopted a two-stage training protocol for AR-LLMs to endow them with order-agnostic abilities without pretraining. Specifically:
>
> **First Stage**. Fine-tune the last layer with target-aware RoPE, adapting next-token prediction to multi-position prediction.
>
> **Second Stage**. Freeze and gradually unlock the last layer during full fine-tuning to stabilize the autoregressive loss.
>
> While effective for stabilizing forward prediction, this method incurs an order-agnostic training tax, with next-token prediction performance dropping from $77.0$% (baseline) to $76.5$% and $74.1$% after the first and second stages, respectively. This likely arises from differences in training objectives and corrupted data incorporation during fine-tuning. Exploring pretraining with order-agnostic modeling could mitigate this issue. We leave this to future work due to the computational constraint.
>
> Please find a more detailed discussion regarding the challenges in training and optimization in the Limitations section of our updated manuscript.
>
> > It would be interesting to compare this ablation with a method from the related work that has a similar computational cost.
>
> Thanks for your thoughtful suggestion! To ensure a fair comparison, we included self-consistency (SC) using $4$ outputs sampled by the next-token prediction baseline. Note that for other related works on iterative refinement, we instead compare the performance gains across different approaches, considering the discrepancies in base models and training data. Below are the results on GSM8K.
>
> | Approach | Base Model | Accuracy (%) | $\Delta$ (Accu) | Speed (tokens per second) | Cost (seconds per sample) |
> | :- | :-: | :-: | :-: | :-: | :-: |
> | Base (Welleck et al. 2023)          | GPT-3 Instruct | $36.8$ | $-$ | $-$ | $-$ |
> | Self-Correct (Welleck et al. 2023) | GPT-3 Instruct |  $45.9$ | $+9.1$ | $-$ | $-$ |
> | Self-Refine (Madaan et al. 2023) | GPT-3 Instruct |  $55.7$ | $+18.9$ | $-$ | $-$ |
> ||
> | SFT (An et al. 2024) | Llama2-7B | $55.0$ | $-$ | $-$ | $-$ |
> | + Learning from mistakes (An et al. 2024) |  Llama2-7B | $57.1$ | $+2.1$ | $-$ | $-$ |
> ||
> | NT | COrAL (Mistral-7B) | $74.1$ |$-$ | $39.7$ | $3.67$ | $-$ | $-$ |
> | SC@$4$                                             | COrAL (Mistral-7B) |   $76.2_{±0.4}$ | $+2.1$ | $37.8$ | $15.50$ |
> | Ours $_\textrm{w/o multi-forward}$ | COrAL (Mistral-7B) |   $78.7$ | $+4.6$ | $14.8$ | $9.81$ |
>
> The results show that our approach (w/o multi-forward) consistently outperforms both the next-token and SC baselines, achieving higher accuracy while consuming less time per sample. Furthermore, compared with other mistake-correction fine-tuning approaches using the base model of the same size, our method achieves a large performance gain (e.g., $+4.6$% compared to $+2.1$% on Llama2-7B).
>
> > comparison to refinement methods from the related work is missing
>
> Our approach introduces a novel framework by converting the output-level sequential refinement into an internal order-agnostic decoding process. Below is a comparison with existing methods:
>
> **Prompt Engineering**. Works such as Self-Refine (Madaan et al. 2023) and Reflexion (Shinn et al., 2024) exploit incorrect attempts in historical data to improve the performance of a frozen LLM. In contrast, our method enables the model to directly correct generated mistakes via backward refinement.
>
> **Verifier Training**. Such approaches (Cobbe et al., 2021; Lightman et al., 2023) train separate models to re-rank outputs. These strategies are orthogonal to our method, which ​​could further enhance COrAL by providing stronger verification mechanisms.
>
> **Mistake-Correction Fine-Tuning**. An et al. (2024) demonstrate that the mistake reasoning data can be directly utilized through a standard fine-tuning approach. However, this approach relies on AR-LLMs and sequential prediction, whereas COrAL introduces a fundamentally new paradigm by enabling mistake correction through backward dependencies.
>
> We have added a detailed discussion of these distinctions in Appendix B.
>
> > pseudo-code for Algorithm 1 is provided without walking through the pseudo-code
>
> We appreciate your comment for elaboration on Algorithm 1. We have included a brief walkthrough of Algorithm 1 in Section 3 (lines 247–253) to clarify its implementation.
>
> > a somewhat non-standard notation for expected values is used. their subscripts seem to be used much like in summations, but usually subscripts at an expected value are used to indicate over which distribution the expectation is taken: e.g., equation (1) and equation (3)
>
> Thank you for pointing this out. We have revised the notation in equations (1) and (3) to align with standard conventions.

---

> ### Author Response · Authors · 2024-11-21
> **Response to Reviewer oenX (2/2)**
>
> > It would be really interesting to check how much performance is lost by starting from a pretrained model as compared to full training a method employing coral from scratch. Do you think that some performance is left on the table because you start from a pretrained model?
>
> Thanks for your insightful suggestions on employing COrAL from scratch through pretraining to finetuning! We agree that pretraining COrAL from scratch could unlock further potential. Our findings on the order-agnostic training tax suggest that **discrepancies between pretraining and fine-tuning objectives can degrade performance**. While our work can be viewed as an initial step to explore the potential to employ order-agnostic modeling for generative language models, we leave it to future work to explore this promising direction of training COrAL from scratch due to computational and resource constraints.
>
> Existing work (e.g., Gloeckle et al., 2024) highlights that **pretraining with multi-token prediction yields more accurate models than fine-tuning alone**. They show that pretraining with multi-token prediction allows the additional heads to be much more accurate than a simple finetuning of a next-token prediction model, thus allowing the models to unlock self-speculative decoding’s full potential. Likewise, Ye et al. (2024) demonstrate the importance of starting from the pretraining stage to learn the skill of error correction, which cannot be acquired by simply applying LoRA finetuning. Following this direction, future work could explore training COrAL from scratch to fully harness its order-agnostic capabilities.
>
> > Figure (2) is a little bit unclear to me. Why are there seemingly different offsets for the refinements and why is there not much visual seperation inbetween forward prediction and refinement?
>
> Thanks for the clarifying question. The offsets are determined by the position of the last fixed token in the sliding decoding block (or the starting position of the current sliding block). Based on both forward and backward dependencies in the generated context, Figure 2 shows that this internal refinement process amends the duplicate "marine" to "organism". Here, the forward prediction and refinement may be close to each other within such a small context window (i.e., $k=3, b=6$). We refer to the case study in arithmetic reasoning in Figure 8 to illustrate how the backward refinement contributes to correcting the wrong tokens generated in forward prediction.
>
>
> > Maybe it would be a good idea to incorporate an application in which this method shines. E.g., by looking into domains that can benefit from the order-agnostic aspect such as protein language modelling.
>
> We appreciate your insight to explore applications that could benefit from the order-agnostic aspect of COrAL! While our current focus is on reasoning tasks to validate COrAL’s ability to capture multiple dependencies efficiently, exploring domains like protein modeling represents an exciting future direction. We hope to investigate COrAL’s specialization and generalizability across broader tasks as part of future work.
>
> ---
> Sean Welleck, Ximing Lu, Peter West, Faeze Brahman, Tianxiao Shen, Daniel Khashabi, Yejin Choi: Generating Sequences by Learning to Self-Correct. ICLR 2023
>
> Aman Madaan, Niket Tandon, Prakhar Gupta, Skyler Hallinan, Luyu Gao, Sarah Wiegreffe, Uri Alon, Nouha Dziri, Shrimai Prabhumoye, Yiming Yang, Shashank Gupta, Bodhisattwa Prasad Majumder, Katherine Hermann, Sean Welleck, Amir Yazdanbakhsh, Peter Clark: Self-Refine: Iterative Refinement with Self-Feedback. NeurIPS 2023
>
> Noah Shinn, Federico Cassano, Ashwin Gopinath, Karthik Narasimhan, Shunyu Yao: Reflexion: language agents with verbal reinforcement learning. NeurIPS 2023
>
> Karl Cobbe, Vineet Kosaraju, Mohammad Bavarian, Mark Chen, Heewoo Jun, Lukasz Kaiser, Matthias Plappert, Jerry Tworek, Jacob Hilton, Reiichiro Nakano, Christopher Hesse, John Schulman: Training Verifiers to Solve Math Word Problems. CoRR abs/2110.14168 (2021)
>
> Hunter Lightman, Vineet Kosaraju, Yuri Burda, Harrison Edwards, Bowen Baker, Teddy Lee, Jan Leike, John Schulman, Ilya Sutskever, Karl Cobbe: Let's Verify Step by Step. ICLR 2024
>
> Shengnan An, Zexiong Ma, Siqi Cai, Zeqi Lin, Nanning Zheng, Jian-Guang Lou, Weizhu Chen: Can LLMs Learn From Mistakes? An Empirical Study on Reasoning Tasks. EMNLP (Findings) 2024: 833-854
>
> Tianle Cai, Yuhong Li, Zhengyang Geng, Hongwu Peng, Jason D. Lee, Deming Chen, Tri Dao: Medusa: Simple LLM Inference Acceleration Framework with Multiple Decoding Heads. ICML 2024
>
> Fabian Gloeckle, Badr Youbi Idrissi, Baptiste Rozière, David Lopez-Paz, Gabriel Synnaeve: Better & Faster Large Language Models via Multi-token Prediction. ICML 2024
>
> Tian Ye, Zicheng Xu,Yuanzhi Li, Zeyuan Allen-Zhu. Physics of language models: Part 2.2, how to learn from mistakes on grade-school math problems, 2024.

---

> > ### Comment · Reviewer_oenX · 2024-11-23
> >
> > Thanks for the thorough response! I increased my score.

---

> > > ### Author Response · Authors · 2024-11-25
> > > **Thanks for the update and valuable feedback**
> > >
> > > Thanks for appreciating our response and for updating the score. We greatly value your feedback and are happy to know that you found the response satisfactory.

---

### Official Review · Reviewer_DAsr · 2024-11-04

**Soundness:** 4
**Presentation:** 4
**Contribution:** 2
**Rating:** 6
**Confidence:** 5

**Summary:**

The paper proposes COrAL(Context-Wise Order-Agnostic Language Modeling), a novel architecture for language modeling that enhances efficiency in iterative refinement, aiming to reduce inference latency in large language models (LLMs). Traditional autoregressive models, which generate text sequentially, struggle with efficiency due to the natural linear time complexity in inference. COrAL incorporates iterative refinement directly into the model, allowing multi-token generation and backward reconstruction within manageable context windows. This order-agnostic approach enables simultaneous forward and backward decoding within sliding context windows, effectively accelerating inference and improving performance on reasoning tasks. Empirical tests show significant improvements in both accuracy and inference speed, demonstrating COrAL's promise in capturing diverse token dependencies without the high latency typical of AR models. However, challenges remain, such as reduced performance in code generation due to output consistency issues, indicating areas for further refinement.

**Strengths:**

- Improved Efficiency and Performance:  COrAL’s order-agnostic framework allows simultaneous forward and backward processing, significantly reducing inference latency compared to traditional autoregressive models. Compared to the ablated baselines, empirical results on datasets like GSM8K and LogiQA demonstrate notable accuracy gains, confirming the model’s effectiveness in complex reasoning tasks.
- Scalably Adaptable from Existing Models: By using context-wise modeling and target-aware positional encoding, COrAL manages to enhance dependency capture without substantially increasing computational resources, making it feasible for deployment in large-scale applications, even with existing large language models with only minor adaptation.

**Weaknesses:**

- Lack of survey of some (maybe kind of obsolete yet important) existing methods: This method resembles Scheduled Sampling in multiple aspects, yet it severely lacks the acknowledgement of this method (no citation nor even mentioning). It shares many ideas and practices with SS, necessitating a deeper analysis on the connection and differences between the method. For example, I'd recommend the authors to emphasize the capability of the proposed method on semi-parallel, refinitive generation, whereas SS was originally only proposed for improvements of performance in sequential generation.
- Lack of deeper discussion on the theoretical insights: I appreciate the authors' awesome work in presenting and delivering the empirical results, but I presume it would appeal the community more if some insightful conclusions can be presented alongside the experiment observations.

**Questions:**

The clarity of the paper is good, it's easy for people to follow generally. I don't have further questions.

---

> ### Author Response · Authors · 2024-11-21
> **Response to Reviewer DAsr**
>
> We appreciate the reviewer's insightful suggestions!
>
> > Lack of survey of some (maybe kind of obsolete yet important) existing methods: This method resembles Scheduled Sampling in multiple aspects, yet it severely lacks the acknowledgement of this method (no citation nor even mentioning). It shares many ideas and practices with SS, necessitating a deeper analysis on the connection and differences between the method
>
> Thanks for bringing up the work on scheduled sampling (Bengio et al. 2015). We acknowledge the similarities and have added the following discussion in Appendix B to address the connections and differences:
>
> **Motivation and Target Problems to Tackle**. Scheduled sampling aims to mitigate the discrepancy between training and inference, while COrAL introduces a generalized framework to model various dependencies within the context. The motivations and target problems are distinct.
>
> **Training: Mixed Training Schemes**. Scheduled sampling gradually transitions from teacher-forcing to self-generated inputs using curriculum learning. In contrast, COrAL decomposes the order-agnostic training into two separate objectives: forward prediction with ground-truth input and backward reconstruction with corrupted input. Inspired by scheduled sampling, future iterations of COrAL could explore curriculum strategies to gradually increase corrupted input ratios, enhancing robustness and stability.
>
> **Inference: Order-Agnostic v.s. Sequential Decoding**. Scheduled sampling is designed for sequential decoding at inference, while COrAL employs blockwise order-agnostic decoding, enabling multi-token forward prediction for speedup and backward refinement for quality improvement.
>
> **Experiment: Impact of Mixing Ratio**. Both methods highlight the importance of balance. Similar to the findings of the scheduled sampling that pure self-generated training performs poorly, we observe that a high corruption ratio (e.g., $0.5$) in COrAL significantly degrades performance, underscoring the need for carefully designed corruption schemes.
>
> > Lack of deeper discussion on the theoretical insights
>
> We appreciate your suggestion to expand on the theoretical aspects of COrAL. While our focus is on the empirical exploration of order-agnostic modeling, we align our findings with theoretical insights from Zhang et al. (2024), which compare autoregressive and masked paradigms. Below, we summarize the relevant theoretical parallels:
>
> **Enhanced Connectivity in Multi-Token Predictions**. From a graph perspective, we consider the co-occurrence matrix of conditional and target text, where the nodes and the edge weights represent the texts and their joint probability, respectively. Zhang et al. (2024) show that superior downstream performance is linked to enhanced connectivity in the co-occurrence matrix of conditional and target text. Likewise, COrAL’s multi-token dependencies improve connectivity compared to AR models, explaining the effectiveness of our ensemble verification policy (Eq. 6). This aligns with our empirical results, where the ensemble verification mechanism (Ours w/o multi-forward) significantly boosts reasoning task performance across various datasets.
>
> **Impact of Aggressive Mask Ratios**. Larger mask ratios cluster samples more effectively in feature space, as demonstrated in Zhang et al. (2024). This theoretically explains the impact of the corruption ratio on model performance in COrAL. This explains why moderate corruption ratios (e.g., $0.125$–$0.25$) in COrAL enhance reconstruction, as shown in Figure 6(b).
>
> **Autoregressive Models Obtain a Smaller Error Compared to Masked Models**. AR models inherently minimize errors more effectively in generation tasks. However, consistency across output distributions can bridge this gap for masked models. This indicates the importance of the consistency of different positions for better masked modeling. Likewise, our ablation study on the corruption granularity in Figure 6(a) demonstrates how to maintain this consistency by balancing corruption piece lengths and the maximum context window size $k=8$, as short corrupted pieces (e.g., $1, 2$) may break up the coherence of the sequence while longer pieces (e.g., $8$) require longer dependencies that may not be available.
>
> ---
> Samy Bengio, Oriol Vinyals, Navdeep Jaitly, Noam Shazeer: Scheduled Sampling for Sequence Prediction with Recurrent Neural Networks. NIPS 2015: 1171-1179
>
> Qi Zhang, Tianqi Du, Haotian Huang, Yifei Wang, Yisen Wang: Look Ahead or Look Around? A Theoretical Comparison Between Autoregressive and Masked Pretraining. ICML 2024

---

> ### Author Response · Authors · 2024-11-25
> **Looking forward to further feedback**
>
> Dear Reviewer DAsr,
>
> Thank you again for your valuable comments and the effort you put into reviewing our work! We have carefully addressed the main concerns in detail and hope you find our responses satisfactory, as other reviewers have. As the discussion phase is about to close, we look forward to hearing any additional feedback you may have. We will be happy to clarify or provide additional details.
>
> Best,
>
> Authors

---

> > ### Comment · Reviewer_DAsr · 2024-11-26
> >
> > I have read the response and would like to keep my scores. Thanks.

---

> > > ### Author Response · Authors · 2024-11-26
> > >
> > > Thank you for reviewing our paper and considering our rebuttal. We sincerely appreciate your valuable feedback and recognition of our contribution.

---

> > > > ### Author Response · Authors · 2024-11-29
> > > >
> > > > Dear Reviewer DAsr,
> > > >
> > > > Thanks again for your valuable feedback and recognition of our contributions.
> > > >
> > > > With the extended discussion period, we would like to engage in further discussions and address any remaining questions or concerns!
> > > >
> > > > Best,
> > > >
> > > > Authors

---

### Official Review · Reviewer_U9ey · 2024-11-04

**Soundness:** 2
**Presentation:** 3
**Contribution:** 2
**Rating:** 3
**Confidence:** 4

**Summary:**

This paper proposes Context-Wise Order-Agnostic Language Modeling (COrAL), which incorporates iterative refinement directly into the LLM architecture while maintaining computational efficiency. Empirical evaluations on reasoning tasks demonstrate that COrAL improves performance and inference speed, and results on code generation indicate a drop in pass rates due to inconsistencies in order-agnostic outputs, highlighting the inherent quality–speed trade-off.

**Strengths:**

- This paper is well-writen and easy to follow.
- The performance on logical reasoning tasks are good.

**Weaknesses:**

- I think this paper is similar to the other type of works, i.e., speculative decoding, what the difference between them?
- The noverty is limited, since the specific ways for iterative refinements, the training methods to learn correction, are borrowed from previous works.
- The significant one: this method seems to only work in specific tasks, the logical reasoning tasks in this paper. However, we always focus on the generalization of current language models, i.e., the competitive on a wide range of tasks.

**Questions:**

- If the way to generate tokens in the first step is different from that in the process of iterative refinements? Are there any better methods to generate draft tokens.

---

> ### Author Response · Authors · 2024-11-21
> **Response to Reviewer U9ey (1/2)**
>
> Thanks for taking the time to review our work. Below, we clarify the contributions and novelty of our work first and address your comments.
>
> ---
> We hope to highlight our research focus first to clarify our main contribution as follows:
> * **Introduction of COrAL**: We present a language modeling approach that unifies denoising with context-wise order-agnostic language modeling, effectively combining the strengths of AR and NAR models.
> * **Development of Blockwise Order-Agnostic Decoding**: We propose an efficient decoding strategy that enables multi-token prediction and backward reconstruction within context windows, enhancing both performance and inference speed.
> * **Application of Target-Aware Positional Encoding**: We employ a generalized Rotary Position Embedding in the Transformer architecture to maintain target-aware positional information without modifying the model's architecture or necessitating extensive pretraining.
> * **Empirical Validation**: We demonstrate through comprehensive experiments that COrAL achieves significant improvements in accuracy and inference speed on reasoning tasks, while also discussing the limitations observed in code generation tasks.
> * Our approach offers a promising direction for developing more efficient and capable large language models by effectively capturing local dependencies within context windows while maintaining computational efficiency.
> ---
>
> > I think this paper is similar to the other types of works, i.e., speculative decoding, what is the difference between them?
>
> Thanks for the clarifying question. Our proposed decoding method, blockwise order-agnostic decoding, differs from speculative decoding in the following key aspects:
>
> **No Separate Draft Model**: The typical speculative decoding approach (Chen et al., 2023; Leviathan et al., 2023) employs a smaller, faster draft model to propose multiple continuations, which the larger target model then verifies and accepts. This inherently adds memory overhead and limits distributional deployment, while our approach leverages the order-agnostic capability of COrAL to generate the draft tokens using the same model, ensuring scalability and efficiency.
>
> **Orthogonal Strategy of Draft Token Generation Compared to Self-Speculative Decoding**: Self-speculative decoding (Zhang et al. 2024) uses the same model for drafting by selectively skipping certain intermediate layers. However, this may take hours to configure and limit interpretability and generalization. COrAL instead uses order-agnostic generation to break sequential dependencies in AR-LLMs, enabling efficient multi-token drafting.
>
> **Quality Improvement via Backward Refinement**: Previous works of speculative decoding mainly focus on inference acceleration with lightweight drafting, while our approach combines speed-up with quality improvements through iterative backward refinement of generated content.
>
> We have clarified these differences in Section 3 (lines 237-245) and have now expanded the discussion in Appendix B to provide further detail.
>
> > The novelty is limited, since the specific ways for iterative refinements, the training methods to learn correction, are borrowed from previous works
>
> We appreciate your concern about the novelty. Our approach introduces a novel framework by converting the output-level sequential refinement into an internal order-agnostic decoding process. Below is a comparison with existing methods:
>
> **Prompt Engineering**. Works such as Self-Refine (Madaan et al. 2023) and Reflexion (Shinn et al., 2024) exploit incorrect attempts in historical data to improve the performance of a frozen LLM. In contrast, our method enables the model to directly correct generated mistakes via backward refinement.
>
> **Verifier Training**. Such approaches (Cobbe et al., 2021; Lightman et al., 2023) train separate models to re-rank outputs. These strategies are orthogonal to our method, which ​​could further enhance COrAL by providing stronger verification mechanisms.
>
> **Mistake-Correction Fine-Tuning**. An et al. (2024) demonstrate that the mistake reasoning data can be directly utilized through a standard fine-tuning approach. However, this approach relies on AR-LLMs and sequential prediction, whereas COrAL introduces a fundamentally new paradigm by enabling mistake correction through backward dependencies.
>
> We have added a detailed discussion of these distinctions in Appendix B.

---

> ### Author Response · Authors · 2024-11-21
> **Response to Reviewer U9ey (2/2)**
>
> > This method seems to only work in specific tasks, the logical reasoning tasks in this paper. However, we always focus on the generalization of current language models, i.e., the competitive on a wide range of tasks.
>
> We appreciate the reviewer’s concern about the generalizability of COrAL.
>
> We would like to highlight that COrAL achieves **significant improvements in accuracy and inference speed on various reasoning tasks**, spanning from arithmetic reasoning, including GSM8K ($+4.6$%) and MATH ($+2.5$%), to logical reasoning containing LogiQA ($+4.0$%) and ReClor ($+1.5$%). Our comprehensive experiments demonstrated the potential of order-agnostic language modeling to enhance reasoning through internal iterative refinement. On the other hand, our extended experiment on code generation also shows the limitations of COrAL in tasks that require strict syntactic coherence. This evaluation **highlights both the strengths and limitations of COrAL, providing valuable insights for future research**.
>
> Due to computational and data constraints, we leave the exploration of other tasks (e.g., instruction following, dialogue, infilling) to future work to further investigate the generalizability and specialty of COrAL.
>
> > If the way to generate tokens in the first step is different from that in the process of iterative refinements, are there any better methods to generate draft tokens?
>
> Besides order-agnostic generation, alternative strategies such as separate draft models (Chen et al., 2023; Leviathan et al., 2023) or self-speculative decoding (Zhang et al., 2024) could be employed. However, these methods primarily focus on inference speed-up and are inherently tied to AR-LLMs, making them **fundamentally different** from COrAL, which integrates multi-token generation and backward refinement within an order-agnostic paradigm. Below is a conceptual comparison:
>
> | Approach | Additional Draft Model (Scalability) | AR-based (Multi-Dependency) | Training |
> | :- | :-: | :-: | :- |
> | SpecDecoding | ✓ ($\downarrow$) | ✓ (✗) | to train the draft model |
> | Self-SpecDecoding | ✗ ($\uparrow$) | ✓ (✗) | to determine the layers to skip |
> | COrAL | ✗ ($\uparrow$) | ✗ (✓) | to learn order-agnostic modeling |
>
> ---
> Charlie Chen, Sebastian Borgeaud, Geoffrey Irving, Jean-Baptiste Lespiau, Laurent Sifre, John Jumper: Accelerating Large Language Model Decoding with Speculative Sampling. CoRR abs/2302.01318 (2023)
>
> Yaniv Leviathan, Matan Kalman, Yossi Matias: Fast Inference from Transformers via Speculative Decoding. ICML 2023: 19274-19286
>
> Jun Zhang, Jue Wang, Huan Li, Lidan Shou, Ke Chen, Gang Chen, Sharad Mehrotra: Draft& Verify: Lossless Large Language Model Acceleration via Self-Speculative Decoding. ACL (1) 2024: 11263-11282
>
> Aman Madaan, Niket Tandon, Prakhar Gupta, Skyler Hallinan, Luyu Gao, Sarah Wiegreffe, Uri Alon, Nouha Dziri, Shrimai Prabhumoye, Yiming Yang, Shashank Gupta, Bodhisattwa Prasad Majumder, Katherine Hermann, Sean Welleck, Amir Yazdanbakhsh, Peter Clark: Self-Refine: Iterative Refinement with Self-Feedback. NeurIPS 2023
>
> Noah Shinn, Federico Cassano, Ashwin Gopinath, Karthik Narasimhan, Shunyu Yao: Reflexion: language agents with verbal reinforcement learning. NeurIPS 2023
>
> Karl Cobbe, Vineet Kosaraju, Mohammad Bavarian, Mark Chen, Heewoo Jun, Lukasz Kaiser, Matthias Plappert, Jerry Tworek, Jacob Hilton, Reiichiro Nakano, Christopher Hesse, John Schulman: Training Verifiers to Solve Math Word Problems. CoRR abs/2110.14168 (2021)
>
> Hunter Lightman, Vineet Kosaraju, Yuri Burda, Harrison Edwards, Bowen Baker, Teddy Lee, Jan Leike, John Schulman, Ilya Sutskever, Karl Cobbe: Let's Verify Step by Step. ICLR 2024
>
> Shengnan An, Zexiong Ma, Siqi Cai, Zeqi Lin, Nanning Zheng, Jian-Guang Lou, Weizhu Chen: Can LLMs Learn From Mistakes? An Empirical Study on Reasoning Tasks. EMNLP (Findings) 2024: 833-854

---

> > ### Comment · Reviewer_U9ey · 2024-11-23
> >
> > Thanks for your response.
> > I do agree with you that the contribution of Blockwise Order-Agnostic Decoding and Target-Aware Positional Encoding for the , COrAL framework. However, I am still concern that whether COrAL actually effectively combine the strengths of AR and NAR models.
> > -  Firstly, according to your title, COrAL is specially designed for efficient iterative refinement, but the results do not demonstrate this, except on LogiQA w/o verifier. Actually, original fully NAR models can achieve >10x speedup, and around 3x with iterative refinements, e.g., CMLM models[1,2]. However, the results of COrAL (the results of ours in Tables) only achieve 1.1x or 1.2x speedup, thus I do not think this is efficient.
> > - Compared with works of speculative decoding [3,4], they can also achieve >2.0x speedup without sacrificing performance, even on code generation tasks. Therefore, I agree with you the difference with speculative decoding, but I can not find the advantages of COrAL.
> > - In my opinion, the strengths of AR models are the better capturing of the target token dependency thanks to their simple autoregressive modeling paradigm, however, the order-agnostic decoding seems to complicates the modeling process and the iterative refinement is also not the strength of AR models. Therefore, I think the framework is more likely to be a tool to transform the pre-trained AR models to support multi-token forward prediction and verification.
> > - Overall, I find the key points of COrAL is fuzzy, which also leads to my concern of COrAL. The authors say that COrAL is designed for overcoming the several challenges of AR models. If the challenges are the slow decoding speed, I think COrAL should at least show advantages (higher speedup) compared with speculative decoding, if the challenges are the sub-optimal sub-optimal performance, COrAL should outperform AR model on the most tasks, not the tradeoff. Therefore, I think the actual challenges that COrAL can solve are those specially under the iterative refinements paradigm for pre-trained AR models, but this
> > paradigm is not the main stream of these models.
> >
> > I keep the score now, but I look forward the further discussion with you.
> >
> > [1] Mask-predict: Parallel decoding of conditional masked language models, EMNLP 2019.
> > [2] Are Bert Family Good Instruction Followers? A Study on Their Potential And Limitations ICLR 2024.
> > [3] Eagle: Speculative sampling requires rethinking feature uncertainty
> > [4] Medusa: Simple llm inference acceleration framework with multiple decoding heads

---

> ### Author Response · Authors · 2024-11-25
> **Further Response to Reviewer U9ey (1/2)**
>
> Thanks for your constructive feedback. We appreciate the opportunity to clarify the contributions and advantages of COrAL. Below, we address each of your points in detail.
>
> > **Efficient Iterative Refinement and Speedup**. According to your title, COrAL is specially designed for efficient iterative refinement, but the results do not demonstrate this, except on LogiQA w/o verifier … thus I do not think this is efficient.
>
> COrAL aims to reduce inference latency in AR-LLMs by performing iterative refinement internally via order-agnostic generation rather than relying on AR-based prompting. Its efficiency stems from multi-token prediction and backward reconstruction, achieving up to $3.9\times$ (GSM8K) over the AR baseline without significant performance degradation.
>
> To further probe this efficiency, we conducted two experiments:
> * **Performance–Speed Trade-offs**. Similar to [1], we illustrate the performance–speed trade-offs of COrAL in Figure 5(a). By increasing the decoding block size, COrAL can **approach the maximum speedup rate about $4\times$ while retaining the generation quality**. Moreover, unlike CMLM models [1,2], which face challenges with varying-length predictions, COrAL **maintains the flexibility of AR decoding to handle variable-length generation**.
> * **Scaling Performance with Iterative Refinement**. As shown in Figure 1, COrAL enables **faster performance scaling than the inference cost** as the iteration time increases. Leveraging backward dependencies, it outperforms forward-only refinement, reaching a higher plateau of accuracy at a reasonable computational cost.
>
> > **Advantages Over Speculative Decoding**. Compared with works of speculative decoding [3,4], they can also achieve >2.0x speedup without sacrificing performance … but I can not find the advantages of COrAL.
>
> COrAL demonstrates distinct advantages over speculative decoding, particularly in reasoning tasks. According to Table 1 in [3], on GSM8K, speculative sampling (EAGLE) achieves speedups up to $3.01\times$ and $2.91\times$ on 7B models Vicuna-7B and Llama2-Chat-7B, respectively. By comparison, COrAL achieves $>3.5\times$ speedup with comparable accuracy ($73.0$%), as shown in Figure 5(a). Additionally, COrAL eliminates the need for a separate draft model, reducing memory overhead.
>
> On the other hand, we acknowledge that COrAL faces limitations on tasks requiring strict syntactic coherence, such as code generation. We attribute this to the discrepancy between pretraining and fine-tuning objectives, which affects the quality of draft tokens in order-agnostic generation. Addressing this limitation via pretraining with order-agnostic modeling remains an important direction for future work (Appendix E). We refer to [4], which explores pretraining with multi-token prediction and highlights the potential benefits of incorporating COrAL into the pretraining stage to enhance its capabilities.
>
> > The strengths of AR models are the better capturing of the target token dependency thanks to their simple autoregressive modeling paradigm … I think the framework is more likely to be a tool to transform the pre-trained AR models to support multi-token forward prediction and verification.
>
> Thanks for the recognition of **COrAL’s contribution as a tool to transform pre-trained AR models for multi-token prediction and verification**. While iterative refinement is not inherently a strength of AR models, the prompting-level refinement paradigm has proven effective for enhancing AR-LLMs on complex tasks [5,6]. COrAL addresses the latency challenges of this paradigm by enabling efficient multi-token prediction and backward reconstruction, thus **extending AR models' capabilities for iterative refinement**.
>
> We also respectfully note that other reviewers have recognized the contributions of COrAL in terms of introducing the novel paradigm to address limitations in AR-LLMs: reviewer `DAsr` highlighted that _"COrAL’s order-agnostic framework allows simultaneous forward and backward processing, significantly reducing inference latency compared to traditional autoregressive models"_, reviewer `oenX` mentioned that we _"propose an interesting paradigm"_ and _"introduce a novel decoding strategy combining autoregressive modelling with ROBERTA-like order agnostic refinement"_, and reviewer `LRiM` emphasized that the _"topic of the paper is interesting"_, where _"transformer-based model do have the problem of slow decoding speed"_.

---

> ### Author Response · Authors · 2024-11-25
> **Further Response to Reviewer U9ey (2/2)**
>
> > **Key Contributions of COrAL**. I find the key points of COrAL is fuzzy … I think the actual challenges that COrAL can solve are those specially under the iterative refinements paradigm for pre-trained AR models, but this paradigm is not the main stream of these models.
>
> Thanks for the clarifying question regarding the key contributions of COrAL. While the AR architecture does not inherently support iterative refinement compared to NAR models, AR modeling becomes the de facto standard for generative language modeling. Existing inference-time scaling methods, often implemented at the application or prompting level, rely heavily on AR decoding. This brings inherent limitations, such as potential inductive biases and high inference latency due to the monotonic dependency in next-token prediction.
>
> As discussed above, **COrAL addresses above challenges in AR-LLMs by introducing efficient iterative refinement tailored for complex tasks**. Our results (e.g., w/o verifier and Figures 1 and 5) demonstrate that COrAL achieves significant speedup on reasoning tasks, such as GSM8K, outperforming speculative sampling [3] on 7B models.
>
> Regarding the sub-optimal performance of AR models, we have also demonstrated that our approach (w/o multi-forward) consistently outperforms both the next-token and self-consistency (SC) baselines, **achieving higher accuracy while consuming less time per sample compared to SC** (See Table 1 in our updated version), indicating the promise of COrAL for reducing computational cost and enhancing performance.
>
> We hope this addresses the reviewer's concerns and we look forward to their response!
>
> ---
>
> [1] Marjan Ghazvininejad, Omer Levy, Yinhan Liu, Luke Zettlemoyer: Mask-Predict: Parallel Decoding of Conditional Masked Language Models. EMNLP/IJCNLP (1) 2019: 6111-6120
>
> [2] Yisheng Xiao, Juntao Li, Zechen Sun, Zechang Li, Qingrong Xia, Xinyu Duan, Zhefeng Wang, Min Zhang: Are Bert Family Good Instruction Followers? A Study on Their Potential And Limitations. ICLR 2024
>
> [3] Yuhui Li, Fangyun Wei, Chao Zhang, Hongyang Zhang: EAGLE: Speculative Sampling Requires Rethinking Feature Uncertainty. ICML 2024
>
> [4] Fabian Gloeckle, Badr Youbi Idrissi, Baptiste Rozière, David Lopez-Paz, Gabriel Synnaeve: Better & Faster Large Language Models via Multi-token Prediction. ICML 2024
>
> [5] Aman Madaan, Niket Tandon, Prakhar Gupta, Skyler Hallinan, Luyu Gao, Sarah Wiegreffe, Uri Alon, Nouha Dziri, Shrimai Prabhumoye, Yiming Yang, Shashank Gupta, Bodhisattwa Prasad Majumder, Katherine Hermann, Sean Welleck, Amir Yazdanbakhsh, Peter Clark: Self-Refine: Iterative Refinement with Self-Feedback. NeurIPS 2023
>
> [6] Noah Shinn, Federico Cassano, Ashwin Gopinath, Karthik Narasimhan, Shunyu Yao: Reflexion: language agents with verbal reinforcement learning. NeurIPS 2023

---

> > ### Comment · Reviewer_U9ey · 2024-11-26
> >
> > Thanks for your response, and it truly addresses part of my concerns. Actually, I think 3 score is indeed low of this paper, but the current form of paper does not reach 5. Since there is no 4 score option, I'm sorry I can only keep my score.

---

> > > ### Author Response · Authors · 2024-11-26
> > > **Thanks for the update and valuable feedback**
> > >
> > > Thanks for engaging in discussion with us. We are glad that our responses have addressed part of your concerns. Below, we outline updates to the manuscript that may further address your concerns:
> > >
> > > * **Introduction (lines 101-102)**. We highlighted our contribution of introducing a tool to transform pre-trained AR models for multi-token prediction and verification, as suggested by the reviewer.
> > >
> > > * **Table 1**. We added a comparison to self-consistency, which has a comparable computational cost as the w/o multi-token variant of our decoding approach. This demonstrates the efficiency of the iterative refinement mechanism in COrAL, achieving higher accuracy while consuming less computation cost.
> > >
> > > * **Appendix A**. We made a conceptual comparison between COrAL, AR, and NAR architectures to highlight the advantages of COrAL, such as variable-length generation, multi-dependency, and efficient iterative refinement.
> > >
> > > * **Appendix E**. We provided extensive comparisons to other inference approaches, including speculative sampling and iterative refinement, to clarify the distinctions and strengths of COrAL.
> > >
> > > * **Limitations**. We expanded the discussion on training challenges, optimization issues, and COrAL’s limitations on tasks requiring strict syntactic coherence. This enhances clarity and provides insights for future directions to explore COrAL’s potential and generalizability with more computational resources.
> > >
> > > We believe the additional experiments, analysis, and discussion have significantly improved the quality and clarity of our submission. We hope these enhancements provide sufficient grounds for reconsideration of the score. Please let us know if you have additional questions or concerns.

---

### Meta-Review · Area_Chair_gvtB · 2024-12-20

**Metareview:**

**Summary:**

The paper introduces COrAL, a framework designed to integrate iterative refinement directly into LLMs while maintaining computational efficiency. COrAL addresses limitations of autoregressive models, such as high inference latency and sequential dependency, by modeling token dependencies within context windows in an order-agnostic manner. It combines forward multi-token prediction and backward reconstruction to enable efficient sliding block-wise decoding, achieving improvements in both accuracy and inference speed.

**Strength:**
- COrAL’s order-agnostic framework enables simultaneous forward and backward processing.
- The method enhances dependency modeling with minimal computational overhead.
- The authors conduct extensive ablation studies, showcasing the robustness of their approach in balancing speed and performance.

**Weakness:**
- The paper lacks discussion and citations for related works.
- While the empirical results are robust, the paper could benefit from deeper theoretical discussions to provide insights beyond experimental observations.
- While COrAL excels in math and logic tasks, it struggles with code generation, limiting its general applicability.

**Additional Comments On Reviewer Discussion:**

After carefully reviewing all discussion threads, I conclude that this paper demonstrates sufficient merit. Three out of four reviewers provided positive feedback, and one reviewer expressed a negative opinion. However, this paper slightly falls short of the acceptance bar for the ICLR conference, especially when compared to the higher ratings received by other submissions.

---

### Decision · Program_Chairs · 2025-01-22

Reject